# Associations Between Dietary Factors and Breast Cancer Risk: A Systematic Review of Evidence from the MENA Region

**DOI:** 10.3390/nu17030394

**Published:** 2025-01-22

**Authors:** Najoua Lamchabbek, Chaimaa Elattabi, Abdellatif Bour, Bernadette Chimera, Saber Boutayeb, Lahcen Belyamani, Elodie Faure, Inge Huybrechts, Mohamed Khalis

**Affiliations:** 1Department of Public Health and Clinical Research, Mohammed VI Center for Research and Innovation, Rabat 10112, Morocco; 2Mohammed VI International School of Public Health, Mohammed VI University of Sciences and Health, Casablanca 82403, Morocco; 3Laboratory of Biology and Health, Department of Biology, Faculty of Sciences, University Ibn Tofail, Kenitra 14000, Morocco; 4International Agency for Research on Cancer, World Health Organization, 69366 Lyon, France; 5Faculty of Medicine, Mohammed VI University of Sciences and Health, Casablanca 43150, Morocco; 6Université Paris-Saclay, UVSQ, Inserm, Gustave Roussy, CESP, 94805 Villejuif, France; 7French Network for Nutrition and Cancer Research (Nacre Network), 78350 Jouy-en-Josas, France; 8Higher Institute of Nursing Professions and Health Techniques, Ministry of Health and Social Protection, Rabat 10000, Morocco

**Keywords:** MENA, breast cancer, dietary factors, nutrition, systematic review

## Abstract

Background: The Middle East and North Africa (MENA) region is witnessing a continuous rise in the incidence of breast cancer (BC). This region is characterized by distinct cultural and lifestyle habits. Despite the importance of diet as a modifiable risk factor for BC, its role in the development of BC within the MENA context has not been extensively studied. This systematic review aims to identify and synthesize existing evidence regarding the effect of different dietary factors on BC risk among women from this region. Methods: We systematically reviewed the scientific literature for observational studies that examined the association between specific dietary factors and the risk of BC in MENA, in accordance with the Preferred Reporting Items for Systematic Reviews and Meta-Analyses (PRISMA) statement. Our comprehensive search included databases such as PubMed, Web of Science, ScienceDirect, and Scopus, identified a total of 18,085 records, of which 65 met our inclusion criteria and were assessed for quality using the National Institute of Health Quality Assessment Tool. Results: The findings of the 65 included studies were categorized into food groups, nutrients, and dietary patterns. Studies in the MENA region have consistently shown that the consumption of fruit and vegetables, fish and seafood, and black tea are associated with a reduced BC risk. In contrast, the intake of milk and white bread is linked to an increased risk. Specific dietary patterns such as the Mediterranean diet, a healthy plant-based diet, dietary antioxidant index, and overall healthy dietary patterns have shown a negative association with BC risk. Conversely, the dietary insulin index and load, dietary glycemic index, dietary inflammatory index, and unhealthy dietary patterns are associated with an increased risk of BC. For the remaining dietary factors, research was too limited or inconsistent to draw conclusions. Conclusions: Our findings highlight the significant role of dietary factors in modulating BC risk among women in the MENA region, an area that faces a notable gap in research on this topic. Further studies are essential to deepen our understanding and develop targeted dietary recommendations for BC prevention in this population.

## 1. Introduction

Breast cancer (BC) is the most prevalent cancer among women globally, according to the latest statistics in Global Cancer Observatory, BC presents over 2.2 million new cases, leading to approximately 0.66 million deaths in 2022, and by 2040, the global burden of BC is predicted to increase to over 3 million new cases, and 1 million deaths [1,2]. The Middle East and North Africa (MENA) region has known a notable social and cultural change over recent decades, subsequently, most MENA countries have adopted more Westernized lifestyles [3]. This shift is contributing to increasing rates of non-communicable disease, including BC, which presents a significant public health challenge in this geographical area with an alarming rise in in both its incidence and mortality rates [4].

Behavioral risk factors, particularly related to diet and lifestyle, have been suggested as major contributors to the burden of BC in the MENA region [5,6]. This region is characterized by a diverse range of dietary patterns, influenced by its unique cultural, socioeconomic, and environmental factors [7]. Traditional diets in this region often feature an abundance of fruit, vegetables, whole grains, and traditional foods [8]. However, with globalization and urbanization, there has been a shift towards industrialized diets, characterized by increased processed foods, sugars, fats, animal products, saturated- and trans-fatty acids, and relatively less vitamins and minerals due to a decline in the consumption of fruit and vegetables [6,7,8].

Understanding the link between diet and BC within the MENA population is essential for developing effective, population-specific prevention strategies. While the landscape continues to evolve with new insights and previous reviews in this subject have primarily focused on the Middle East [5,6,9], this systematic review distinguishes itself by broadening the focus to include the entire MENA, and it updates and expands the scope to capture the full diversity of dietary patterns, food, and nutrients intake across this region. To the best of our knowledge, this is the first systematic review to examine and provide a comprehensive overview of the association between dietary components, dietary patterns, and BC risk specifically within women from MENA countries.

## 2. Materials and Methods

The present systematic review was carried out according to Preferred Reporting Items for Systematic Reviews (PRISMA) guidelines.

### 2.1. Protocol and Registration

This systematic review was registered in the database of the international prospective register of systematic reviews PROSPERO (www.crd.york.ac.uk/PROSPERO (accessed on 10 July 2024)) as CRD42022319740.

### 2.2. Data Sources and Search Strategy

We performed a standard literature search, without restriction on the time, of PubMed, Web of Science, Science direct, and Scopus using both controlled vocabulary and keywords. We checked also manually reference lists from selected articles to identify additional relevant articles. A systematic literature search was undertaken initially on 25 December 2022, and later updated on 2 December 2024. The search string included the following terms: Breast cancer/neoplasm/tumor, diet/nutrition, search terms related to MENA countries’ names. The detailed keywords and search strategy used for each database is detailed in Appendix A.

### 2.3. Inclusion and Exclusion Criteria

In this systematic review, articles were eligible for inclusion if they met the following criteria: they reported findings from primary research studies conducted among females with BC in the MENA region, assessed dietary factors influencing BC risk, and were published in French or English. There were no restrictions on the study design or the publication date. Studies without abstracts or full text were excluded, as were studies that mixed male and female BC results or had insufficient data or unclear methodology.

### 2.4. Study Selection and Data Collection Process

The selection of articles was conducted using a thorough process involving the title, abstract, and full text of each article. We used Zotero (version 6.0.36) reference manager software to organize and facilitate the selection process [10]. Eligible articles were identified using the PRISMA flow diagram. Two authors (NL and CE) independently screened all titles and abstracts identified through the search, excluding those clearly irrelevant to the topic. Full texts of potentially eligible papers were then reviewed based on the inclusion criteria for this review. They cross-matched the results to identify and resolve any inconsistencies. Any disagreements during the screening were resolved through discussion and consensus.

### 2.5. Data Extraction

Data were extracted from the included studies using a standardized data extraction form. The following information was collected: author(s), year of publication, country of origin, characteristics of the study population, study design, dietary assessment methods used, dietary factors attributable to BC risk, adjustment for confounding, and main findings related to the association between dietary factors and BC risk variables Appendix A.

### 2.6. Quality Assessment

The quality and risk of bias of the included studies were evaluated using the “National Institute of Health (NIH) Quality Assessment Tool for Case-Control Studies, which is a standardized framework designed to assess the internal validity and methodological rigor of case-control studies” [11]. Studies were evaluated based on criteria such as representativeness of the study population, the research question or objective, and the justification for the sample size etc. (Appendix A).

## 3. Results

### 3.1. The Process of Selecting Studies for Inclusion and the Characteristics of the Studies

A total of 18,085 records were identified through the initial database search. After removing duplicates and screening based on titles and abstracts, 84 articles were assessed for eligibility. Through a rigorous selection process, 65 relevant articles were included for data extraction, as presented in Figure 1. All the included studies were case-control studies conducted among women from MENA countries, with no cohort studies identified on this topic in this region. These studies were published between 2004 and 2024, using a range of dietary assessment methods, with the majority utilizing FFQ (Food Frequency Questionnaire), with varying levels of details and differing numbers of food items. The detailed characteristics of the included articles are presented in Appendix A. Most of these studies were conducted in Iran, followed by Jordan, then Turkey and Saudi Arabia, with only one publication each from Morocco, Iraq, and Algeria (Appendix A).

The findings of the present systematic review were grouped into food groups, nutrients, and dietary patterns. Table 1 illustrates the number of studies that reported significant positive, significant negative, or non-significant associations between these dietary components or patterns and BC in the MENA region, providing an overview of the observed relationships. 

Table 2, Table 3, Table 4, Table 5, Table 6, Table 7, Table 8, Table 9, Table 10 and Table 11 provide detailed information on the included studies, with each table focusing on specific category of dietary factors to facilitate comparison across studies. 

### 3.2. Fruit and Vegetable Consumption

Table 2 bellow summarize eleven studies that examined the association between BC and the consumption of fruit and vegetables among women in the MENA region. The total number of participants was 5528 and the sample sizes ranged from 130 to 1010 participants. The findings of these studies varied, with some suggesting a protective effect of high fruit and vegetable intake against BC, while others found no significant association.

The majority of studies highlighted a potential protective role of high fruit and vegetable intake. Al Qadire et al. in a study involving 823 participants, of whom 405 had cases of BC, and Laamiri et al. and Zahedi et al. in a study involving 300 participants, of whom 150 had cases of BC, found that increased fruit and vegetable consumption was associated with a reduced risk of BC among women [12,13,14]. Similarly, Azzeh et al. in a study of 432 participants (214 BC cases) [15], Marzbani et al. in a study of 620 participants (212BC cases) [16], and Hosseinzadeh et al. in a study of 620 participants (212 BC case) [17], reported that a diet containing a high fruit and vegetable intake was linked to a lower BC risk.

However, a study of 450 participants (225 BC cases), conducted by Ahmadnia et al. in Iran’s Guilan Province, suggested that only vegetable consumption was significantly linked to a reduced risk of BC, while fruit consumption showed no significant effect [18]. On the other hand, Fararouei et al., in a study involving 1010 participants (505 BC cases), similarly concluded that BC patients consumed fewer fruit and vegetables than the control group, though the association with BC risk was greater for fruit intake compared to that of vegetables [19].

Conversely, some studies could not find any significant association between either fruit or vegetable consumption and BC risk. For example, a study involving 300 participants (150 BC cases) conducted by Safabakhsh et al. concluded that a higher consumption of fruit and vegetables was not significantly associated with the risk of BC among Iranian women [20]. This lack of significant association was also observed in studies conducted in Turkey by Toklu et al. among 130 participants (65 BC cases) and Ceber et al. among 243 participants (123 BC cases), which reported that dietary habits, including fruit and vegetable intake, did not significantly impact BC risk [21,22].

**Table 2 nutrients-17-00394-t002:** Findings of studies assessing the association between fruit and vegetable intake, and the risk of BC in the MENA region.

Reference	Country	Study Design	Sample Size (Cases/Controls)	Adjustment for Confounding Variables	Results OR (95% CI)
Al Qadire et al. (2018) [12]	Jordan	Case control	823 (405/418)	Physical activity, calcium intake and breast cancer self-examination, BMI, total caloric intake. employment, income, education level, family size, smoking, physical activity, cancer background, menstruation start, and contraceptive usage.	Fruit and vegetables: 0.79 (0.53–0.96) *
Azzeh et al. (2022) [15]	Saudi Arabia	Case control	432 (214/218)	Age, BMI, employment, family income, education, family size, marital status, physical activity, smoking, family history of BC, other health problems, contraceptive use, age at menarche, age at menopause, and breastfeeding duration.	Fruit and vegetables (s/d) 3–5 vs. >1: 0.161 (0.043–0.605) * Leafy vegetables (s/w): NS
Marzbani et al. (2019) [17]	Iran	Case control	620 (212/408)	Age, education level, and BMI.	Ref: daily vegetables 2–3 s/w: Crude: 2.5 (1.5, 3.9) */M1: 1.7 (1.0, 2.9) * 2–3 s/m: Crude: 2.8 (1.9, 4.2) */M1: 2.8 (1.7, 4.5) * Fruit 2–3 s/w: Crude: 2.3 (1.5, 3.5) * 2–3 s/m: Crude: NS
Ceber et al. (2005) [21]	Turkey	Case control	243 (123/120)	NR	Rarely vs. almost daily Vegetables: NS Fruit: NS
Fararouei et al. (2018) [19]	Iran	Case control	1010 (505/505)	Age, education, BMI, occupation, age at first marriage, history of breast disease, family history of BC, oral contraceptive usage, intensity of physical activity, smoking, passive smoker, and dietary habits factors	Fruit (0–4 vs. 8–10 p/w): 1.96 (1.07–3.82) *
Toklu et al. (2018) [22]	Turkey	Case control	130 (65/65)	Parity, passive smoking, cooking methods, MI.	Fruit: NS
Hosseinzadeh et al. (2014) [17]	Iran	Case control	420 (140/280)	Education level, menopause status, oral contraceptive use, Migration, stress, passive smoking, abortion, and breastfeeding.	Fruit–vegetables: 0.22 (0.12–0.39) *
Zahedi et al. (2015) [14]	Iran	Case control	300 (150/150)	NR	Vegetables Univariate: (1≥ vs. 4 ≤ t/w): 5.04 (2.27–9.35) * Multivariate: (1≥ vs. 4 ≤ t/w): 5.78 (1.66–20.08) * Fruit Univariate: (7> vs. 14 ≤ t/w): 3.69 (2.25–6.05) * Multivariate:(7> vs. 14 ≤ t/w): 3.60 (1.33–9.73) *
Safabakhsh et al. (2021) [20]	Iran	Case control	300 (150/150)	M1: BMI, physical activity, energy intake, education. marital status, and menopause status, socioeconomic status, alcohol use, smoking, vitamin supplements use and medication use, comorbidities, length of oral contraceptives use, hormone replacement therapy, age at menarche, time since menopause in postmenopausal women, weight at age 18 years old, number of children, breast feeding ages, and family history of BC.	T1 vs. T3 Total vegetables and fruit g/d: NS Total vegetables g/d: NS Cruciferous vegetables g/d: NS Green leafy vegetables g/d: NS Dark yellow vegetables g/d: NS Other vegetables g/d: NS Total fruit g/d: NS Berry fruit g/d Crude: 0.33 (0.18–0.59) */M1: 0.36 (0.09–1.37) * Citrus fruit g/d Crude: 2.16 (1.22–3.80) */M1: NS Other fruit g/d: NS
Ahmadnia et al. (2016) [18]	Iran	Case control	450 (225/225)	NR	Vegetables (promise) <3 vs. >5: 0.5 (0.3–0.9) * 3–5 vs. >5: 0.2 (0.1–0.3) * Fruit (promise) <2 vs. >4: NS 2–4 vs. >4: NS
Laamiri et al. (2014) [13]	Morocco	Case control	800 (400/400)	Age, body mass index, and menopausal status	Fruit (t/w) 0.001 (0.00–0.004) * Vegetable (t/w) 0.82 (0.22–3.08) *

OR (95% CI): odds ratio (confidence interval), significance threshold *p* < 0.05 *, these results were statistically significant, NS: no significant association; NR: not reported; s/d: servings/day, reference: ref. s/w: serving/week, s/m: serving/month, p/w: portion/week, t/w: times/week, g/d: gram/day.

### 3.3. Milk and Dairy Products

Table 3 presents mixed findings from the ten studies included in the present systematic review, investigating the association between BC risk and the intake of dairy products in the MENA region. The consumption of dairy products has been examined for its potential impact on BC risk. A case-control study conducted by Azzeh et al. (2022) in the Makkah region of Saudi Arabia involved 432 participants (214 BC cases) and found that consuming 1–5 servings of dairy products per day was associated with a reduced risk of BC [15]. Similarly, a case-control study of 375 participants (100 cases) conducted in Iran by Bahadoran et al. suggested a protective role of total dairy, high-fat dairy, low-fat dairy, and fermented dairy products against BC, though non-fermented dairy did not show a significant association [23].

In contrast, Dashti et al., in a study of 1050 participants (350 cases), reported that high-fat dairy was positively associated with BC risk, while low-fat dairy had a protective effect [24]. However, other studies with a simple size ranging from 379 to 800 participants have failed to find any significant association between overall dairy consumption and BC risk [13,16,18,25,26].

The four included studies consistently indicated that total milk consumption might increase the risk of BC. Zahedi et al. in a case-control study conducted in Iran with 300 women (150 BC cases) reported a significant association between higher milk consumption and an increased risk of BC. Similarly, Maliou et al., in a study involving 379 Algerian women (184 BC cases), also observed a significant elevated BC risk with greater milk intake. Furthermore, Mobarakeh et al., in a smaller study of 93 (53 BC cases) Iranian women, found that milk consumption, particularly high-fat milk, was linked to an increased BC risk [14,25,27]. However, Laamiri et al., in a study of 800 Moroccan women (400 BC cases), could not identify any significant association between milk consumption and BC risk [13].

For yogurt intake, four studies assessed its effect on BC risk in the MENA region. One study conducted by Zahedi et al., in Yazd Province, found that women who consumed more than two glasses of yogurt per week had a lower risk of developing BC [14]. However, the three other case-control studies conducted by Dashti et al., Ghalib et al., and Mobarakeh et al. reported contrasting findings, indicating no significant association between yogurt intake and BC risk [24,26,27].

Cheese consumption has been a focal point in three studies. Most of them indicate no significant association between the consumption of various types of cheese and BC risk. Ali Ghalib et al. and Dashti et al. found no overall association between cheese consumption and BC risk in their respective case-control studies conducted in Iraq and Iran [24,26]. Furthermore, Maliou et al. reported that the intake of dairy products, including cheese, did not significantly affect BC risk, except for fresh cheese, which appeared to have a protective effect among Algerian women [25].

**Table 3 nutrients-17-00394-t003:** Findings of MENA studies assessing the association between dairy products intake and the risk of BC.

Reference	Country	Study Design	Sample Size (Cases/Controls)	Adjustment For Confounding Variables	Results OR (95% CI)
Maliou et al. (2017) [25]	Algeria	Case control	379 (184/195)	M1: Age. M2: Age, body mass index, marital status, educational level, occupation, menopausal status, age at menarche, age at menopause, parity, age at first birth and breastfeeding duration, history of BC in first- and second-degree relatives, history of benign breast disease, smoking status, physical activity, oral contraceptive use, and hormone replacement therapy use.	Q1 vs. Q4 Dairy products Overall: NS Premenopausal: NS Postmenopausal: NS Total milk Overall: M1: 2.22 (1.20–4.11) * M2: 2.61 (1.32–5.16) * Premenopausal: NS Postmenopausal: NS T1 vs. T3 Spread, soft and hard cheese Overall: NS Premenopausal: NS Postmenopausal: NS Fresh cheese M1: 0.29 (0.15–0.56) * M2: 0.24 (0.12–0.51) * Premenopausal: NS Postmenopausal: NS
Ahmadnia et al. (2016) [18]	Iran	Case control	450 (225/225)	NR	Dairy products (gl/w): <2 vs. >3: NS 2–3 vs. >3: NS
Azzeh et al. (2022) [15]	Saudi Arabia	Case control	432 (214/218)	Age, BMI, employment, family income, education, family size, marital status, physical activity, smoking, family history of BC, other health problems, contraceptive use, age at menarche, age at menopause, and breastfeeding duration.	Dairy products (s/d) <1 vs. 1–2: 0.178 (0.037–0.859) * <1 vs. 3–5: 0.038 (0.004–0.372) *
Bahadoran et al. (2014) [23]	Iran	Case control	375 (100/175)	M1: Age. M2: Age at menarche, age at firs pregnancy, number of full pregnancies, smoking, use of oral contraceptive, and use of bra. M3: Body mass index and life satisfaction. M4: Menopause status, family history of BC, physical activity, energy intake, and energy density of the diet.	Q1 vs. Q4 Total dairy intake M1: 0.29 (0.14–0.62) * M2: 0.10 (0.04–0.27) * M3: 0.08 (0.03–0.23) * M4: 0.14 (0.04–0.38) * Low-fat dairy M1: 0.14 (0.04–0.38) * M2: 0.25 (0.11–0.54) * M3: 0.08 (0.03–0.22) * M4: 0.10 (0.03–0.34) * High-fat dairy M1: 0.51 (0.26–1.02) * M2: NS M3: 0.57 (0.21–1.56) * M4: 0.54 (0.18–1.6) * Fermented dairy M1: 0.27 (0.12–0.59) * M2: 0.25 (0.11–0.57) * M3: 0.08 (0.03–0.22) * M4: 0.06 (0.02–0.19) * Non-fermented dairy: NS
Dashti et al. (2022) [24]	Iran	Case control	1050 (350/700)	M1: Age and energy intake. M2: Age, energy intake, and other patient backgrounds (such as marriage, place of residence, educational status, alcohol consumption, smoking, menopausal status, family history of BC, history of disease, physical and breastfeeding history, supplement use, breastfeeding history. M3: age, energy intake, other patient backgrounds, and food intake. M4: BMI.	Q1 vs. Q4 Low fat dairy Crude: 0.16 (0.10–0.23) * M1: 0.10 (0.06–0.16) * M2: 0.08 (0.05–0.13) * M3: 0.06 (0.04–0.11) * M4: 0.07 (0.04–0.13) * High-fat dairy Crude: 10.86 (7.0116.83) * M1: 7.98 (5.07–12.57) * M2: 10.17 (6.22–16.62) * M3: 7.87 (4.55–13.60) * M4: 8.62 (4.78–15.55) * T1 vs. T3 Total milk Crude: 2.59 (1.89–3.53) * M1: 1.94 (1.39–2.70) * M2: 2.08 (1.47–16.95) * M3: 1.01 (0.68–1.50) * M4: 1.76 (1.16–2.65) * Total yogurt: NS Total cheese: NS
Ghalib et al. (2019) [26]	Iraq	Case control	676 (338/338)	Age, marriage, residency, education, occupation, economic status, and menopause status.	Ref (Yearly/never) Dairy product: NS Yogurt: NS Cheese: NS
Laamiri et al. (2014) [13]	Morocco	Case control	800 (400/400)	Age, body mass index, and menopausal status.	Milk (t/w): NS Dairy products (t/w): NS
Zahedi et al. (2015) [14]	Iran	Case control	300 (150/150)	NR	Total milk univariate: (<1 vs. 3 gl/w): 3.53 (1.47–7.15) * Yogurt Univariate: (<1 vs. ≥1.5 gl/w): 2.95 (1.7–5.11) * Multivariate: (<1 vs. ≥1.5 gl/w): 2.57 (1.01–6.55) *
Marzbani et al. (2019) [16]	Iran	Case control	620 (212/408)	Age, education level, and BMI.	Dairy consumption (ref: Daily) 2–3 s/w: NS 2–3 s/m: NS
Mobarakeh et al. (2014) [27]	Iran	Case control	93 (53/40)	Age, BMI, and education	High-fat milk: 7.45 (2.19–138.98) * High-fat yogurt: NS High-fat cheese: 6.88 (1.44–32.77) *

OR (95% CI): odds ratio (confidence interval), significance threshold *p* < 0.05 *, these results were statistically significant, NS: no significant association; NR: not reported, reference: ref., Q: quartile, s/d: serving /day, s/w: serving/week, s/m: serving/month, t/w: times/week, gl/w: glass/week.

### 3.4. Meat and Meat Products

Table 4 summarizes four studies investigating the association between meat consumption and BC risk in the MENA region, with sample sizes ranging from 243 to 1010 women. All these studies assessed meat intake without considering cooking methods. For red meat, Laamiri et al. and Fararouei et al. in their case-control studies, including 800 participants (400 breast cancer cases) and 1010 participants (505 breast cancer cases), respectively, reported a positive association with an increased risk of BC [13,19]. While Azzeh et al. and Ceber et al. in their studies involving 432 participants (214 breast cancer cases) and 243 participants (123 breast cancer cases), respectively, found no significant link [15,21]. Processed meat consumption has been linked to a higher risk of BC in one study [13]. When it comes to poultry intake, the evidence remains inconclusive. While one study found a negative association between poultry consumption and BC risk [13], other studies couldn’t find any significant associations [15,21].

**Table 4 nutrients-17-00394-t004:** Findings of studies assessing the association between the consumption of meat and meat products and the risk of BC in the MENA region.

Reference	Country	Study Design	Sample Size (Cases/Controls)	Adjustment for Confounding Variables	Results OR (95% CI)
Azzeh et al. (2022) [15]	Saudi Arabia	Case control	432 (214/218)	Age, BMI, employment, family income, education, family size, marital status, physical activity, smoking, family history of BC, other health problems, contraceptive use, age at menarche, age at menopause, and breastfeeding duration.	1>: ref. Red and processed meat (s/d): NS Poultry (s/d): NS
Ceber et al. (2005) [21]	Turkey	Case control	243 (123/120)	NR	Rarely vs. almost daily Red meat (Meat, beef, and lamb): NS Poultry: NS
Fararouei et al. (2018) [19]	Iran	Case control	1010 (505/505)	Age, education, body mass index, occupation, age at first marriage, history of breast disease, family history of BC, oral contraceptive usage, intensity of physical activity, smoking, passive smoker, and dietary habit factors.	Red meat (8–10 vs. 0–2 p/w): 1.15 (1.04–1.28) *
Laamiri et al. (2014) [13]	Morocco	Case control	800 (400/400)	Age, body mass index, and menopausal status.	Red meat (t/w) 4.61 (2.26–9.44) * Processed meat (t/w) 9.78 (4.73–20.24) * Poultry (t/w) 0.61 (0.46–0.81) *

OR (95% CI): odds ratio (confidence interval), significance threshold *p* < 0.05: *, these results were statistically significant, NS: no significant association; NR: not reported; s/d: servings/day, reference: ref., p/w: portion/week, t/w: times/week.

### 3.5. Fish and Shellfish

Five studies conducted in the MENA region investigated the association between fish and seafood consumption and breast cancer (BC) risk, involving sample sizes ranging from 243 to 1010 women, and summarized in Table 5. These studies reported mixed findings. Studies by Azzeh et al. in Saudi Arabia, Laamiri et al. in Morocco, and Ali Ghalib et al. in Iraq reported a potentially protective association, indicating that a higher consumption of fish and seafood may significantly lower the risk of developing BC [13,15,26]. Conversely, a study by Fararouei et al. in Iran identified a positive association between fish intake and BC risk, suggesting that higher consumption of fish could potentially increase the likelihood of developing BC [19]. Meanwhile, research by Ceber et al. in Turkey found no significant association between fish and seafood consumption and BC risk [21].

**Table 5 nutrients-17-00394-t005:** Findings of the studies assessing the association between the intake of fish and shellfish and the risk of BC in the MENA region.

Reference	Country	Study Design	Sample Size (Cases/Controls)	Adjustment for Confounding Variables	Results OR (95% CI)
Azzeh et al. (2022) [15]	Saudi Arabia	Case control	432 (214/218)	Age, BMI, employment, family income, education, family size, marital status, physical activity, smoking, family history of BC, other health problems, contraceptive use, age at menarche, age at menopause, and breastfeeding duration.	Fish and seafood (s/d) 1–2: 0.211 (0.82–0.545) * 3–5: 0.072 (0.202–0.265) *
Ceber et al. (2005) [21]	Turkey	Case control	243 (123/120)	NR	Rarely vs. almost daily fish/seafood: NS
Fararouei et al. (2018) [19]	Iran	Case control	1010 (505/505)	M1: Age, education, BMI, occupation, age at first marriage, history of breast disease, family history of BC, oral contraceptive usage, intensity of physical activity, smoking, passive smoker, and dietary habits.	Fish (>8 p/w vs. 0–2 p/w): 1.55 (1.12–2.76) *
Ali Ghalib et al. (2019) [26]	Iraq	Case control	676 (338/338)	Age, marriage, residency, education, occupation, economic status, menopause status.	Fish (yearly or never) vs. (≥1 t/w): 0.676 (0.478–0.954) *
Laamiri et al. (2014) [13]	Morocco	Case control	800 (400/400)	Age, body mass index, and menopausal status.	Fish (t/w): never vs. less than once a week 0.07 (0.02–0.24) *

OR (95% CI): odds ratio (confidence interval), significance threshold *p* < 0.05: *, these results were statistically significant, NS: no significant association; NR: not reported; s/d: servings/day, p/w: portion/week, t/w: times/week.

### 3.6. Bread and Cereal Products

The review included seven studies examining the relationship between bread and cereal consumption and BC risk in the MENA region, with sample sizes ranging from 400 to 800 participants as summarized in Table 6. Among these, two studies reported a positive association between white bread consumption and an increased risk of BC among women in Jordan and Iran [28,29]. In contrast, the consumption of whole wheat bread and cereals showed mixed results in five studies. Some studies found a negative association [18,28], while others reported no significant association [13,28,29].

**Table 6 nutrients-17-00394-t006:** Findings of studies assessing the association between the intake of bread and cereal products and the risk of BC in the MENA region.

Reference	Country	Study Design	Sample Size (Cases/Controls)	Adjustment for Confounding Variables	Results OR (95% CI)
Hammad et al. (2020) [28]	Jordan	Case control	400 (200/200)	Age, marital status, education, income, BMI, total physical activity, smoking, and member of family diagnosed with cancer.	<1: Ref White Bread >1 daily: 1.93 (0.81–4.63) * Whole wheat bread >1 daily: 0.51 (0.25–1.07) * Breakfast cereals: NS
Laamiri et al. (2014) [13]	Morocco	Case control	800 (400/400)	Age, body mass index, and menopausal status.	Cereal (t/w): NS
Tajaddini et al. (2015) [29]	Iran	Case control	615 (306/309)	M1: Age at menopause, total calorie, parity, and BMI	<1: Ref Consumers vs. non-consumers White bread Crude: 1.90 (1.40–2.80) * M1: 1.46 (0.94–2.26) * Whole-wheat bread: NS Cooked cereals (Pasta): NS Rice, white: NS
Ahmadnia et al. (2016) [18]	Iran	Case control	450 (225/225)	NR	Bread and cereals (promise): 6> vs. >11: 0.4 (0.2–0.8) *

OR (95% CI): odds ratio (confidence interval), significance threshold *p* < 0.05: *, these results were statistically significant, NS: no significant association; NR: not reported; reference: ref., t/w: times/week.

### 3.7. Non-Alcoholic Beverages

A total of three studies in the MENA region have examined the relationship between beverage consumption and BC risk, as summarized in Table 7. Ali Ghalib et al. conducted a case-control study in Iraq, included 338 BC cases and 338 controls, and found that daily consumption of black tea more than three times significantly reduced the risk of BC [26]. Similarly, Azzeh et al. [15], in a study of 432 women (214 BC case) from Saudi Arabia, suggested a possible protective effect of black tea consumption, with 1–5 cups per day associated with significantly lower BC risks. Additionally, coffee intake showed a dose-response relationship, with higher consumption associated with lower BC risk. Moreover, Marzbani et al. conducted a case-control study involving 620 Iranian women (212 BC cases) and found that unfavorable dietary patterns, including high consumption of soft drinks and industrially produced juices, were associated with an increased risk of BC [16].

**Table 7 nutrients-17-00394-t007:** Findings of studies assessing the association between the intake of non-alcoholic beverages and the risk of BC in the MENA region.

Reference	Country	Study Design	Sample Size (Cases/Controls)	Adjustment for Confounding Variables	Results OR (95% CI)
Ali Ghalib et al. (2019) [26]	Iraq	Case control	676 (338/338)	Age, marriage, residency, education, occupation, economic status, and menopause status.	Black Tea: >3 t/w: 0.314 (0.144–0.683) *
Azzeh et al. (2022) [15]	Saudi Arabia	Case control	432 (214/218)	Age, BMI, employment, family income, education, family size, marital status, physical activity, smoking, family history of BC, other health problems, contraceptive use, age at menarche, age at menopause, and breastfeeding duration.	Black tea (c/d) 1–2: 0.06 (0.01–0.371) * 3–5: 0.083 (0.009–0.395) * Coffee (c/d) 1–2: 0.159 (0.031–0.812) * 3–5: 0.083 (0.013–0.544) * >5: 0.144 (0.028–0.736) *
Marzbani et al. (2019) [16]	Iran	Case control	620 (212/408)	M1: Age, education level, and BMI.	Unfavorable vs. Favorable Soft drinks Crude: 3.9 (2.7, 5.5) M1: 2.8 (1.9, 4.3) Industrially produced juices Crude: 2.5 (1.5, 3.9) * M1: 1.7 (1.0, 2.9) *

OR (95% CI): odds ratio (confidence interval), significance threshold *p* < 0.05: *, these results were statistically significant, c/d: cups/day, t/w: times/week.

### 3.8. Dietary Fats and Oils

The relationship between dietary fats and oils and BC risk has been extensively studied in the MENA region, revealing diverse findings across different studies. For instance, research has included sample sizes ranging from small group of 80 participants to larger group with 823 participants as described in Table 8.

Marzbani et al. conducted a study in Iran and reported a significant positive association between overall fats and oils consumption and BC risk [16]. Similarly, in Saudi Arabia, Alothaimeen et al. reported that higher intake of total fat was significantly associated with an increased BC risk. Additionally, higher levels of triglycerides, polyunsaturated fats, and cholesterol were also linked to an elevated BC risk [30]. Moreover, Zahedi et al. identified significant positive associations between BC risk and the consumption of solid oils and animal oils in univariate analyses [14].

In contrast, Gholamalizadeh et al. found no significant associations between BC risk and total fat, cholesterol, monounsaturated fats (MUFAs), omega-3 fatty acids, or omega-6 fatty acids in their study in Iran [31]. On the other hand, studies by Al Qadire et al., in Jordan, and Ceber et al. in Turkey could not find any significant associations between dietary fat intake and BC risk [12,21]. Similarly, Toklu et al. reported no significant association with saturated fatty acid intake but suggested a significant association with the use of olive oil [22].

**Table 8 nutrients-17-00394-t008:** Findings of the included studies about the association between the intake of dietary fats and oils and the risk of BC in the MENA region.

Reference	Country	Study Design	Sample Size (Cases/Controls)	Adjustment for Confounding Variables	Results OR (95% CI)
Marzbani et al. (2019) [16]	Iran	Case control	620 (212/408)	M1: Age, education level, and BMI.	Fats and oils (ref: Favorable) Unfavorable: Crude: 2.1 (1.4–3.0) * M1: 1.9 (1.3–3.0) *
Alothaimeen et al. (2004) [30]	Saudi Arabia	Case control	100 (499/498	M1: Age and province M2: Age, nationality, province, menopause, and triglycerides.	Q1 vs. Q4 Total fat (g) Crude: 0.57 (0.40–0.81) * M1: NS M2: 2.67 (1.47–4.83) * Triglycerides (mM/L) Crude: 2.90 (1.79–4.81) * M2: 2.16 (1.21–3.88) * Polyunsaturated fat (g) Crude: 0.62 (0.43–0.88) * M2: 2.43 (1.36–4.34) * Cholesterol (mg) Crude: 0.68 (0.48–0.97) * M2: 1.88 (1.03–3.44) *
Shafie et al. (2023) [32]	Iran	Case control	360 (120/240)	Age, body mass index, energy intake, number of pregnancies, breastfeeding duration, number of abortion cases, family history of breast cancer, and physical activity.	w-3/w-6: NS
Gholamalizadeh et al. (2021) [31]	Iran	Case control	540 (180/360)	M1: Age and BMI. M2: The level of using alcohol drinks, smoking, physical activity, calorie intake, protein intake, and carbohydrate intake.	Total fat: NS Cholesterol: NS MUFAs: NS Omega-3 fatty acids: NS Omega-6 fatty acids M1: 5.429 (2.5–11.79) * M2: 3.398 (1.6–8.4) *
Zahedi et al. (2015) [14]	Iran	Case control	300 (150/150)	NR	(Ref: Liquid oil) Univariate: Solid oil: 8.77 (4.76–16.15) * Animal oil: 3.02 (1.28–7.12) * Multivariate: Solid oil: 9.09 (3.12–26.46) * Animal oil: 4.28 (1.2–19.22) *
Mobarakeh et al. (2014) [27]	Iran	Case control	93 (53/40)	Age, BMI, and education.	Use of olive/frying/liquid oils for cooking: 0.03 (0.005–0.307) * Use of frying oils: NS
Al Qadire et al. (2018) [12]	Jordan	Case control	823 (405/418)	Physical activity, calcium intake, and breast cancer self-examination, BMI, total caloric intake, employment, income, education level, family size, smoking, physical activity, cancer background, menstruation start, and contraceptives usage.	Dietary fat: NS
Alim et al. (2016) [33]	Turkey	Case control	80 (40/40)	Menarche age, age at first birth, Number of children, menopause age, energy, BMI.	Saturated fatty acid: NS
Toklu et al. (2018) [22]	Turkey	Case control	130 (65/65)	Parity, passive smoking, cooking methods, and BMI.	Use of olive oil (t/w): Never rare, 1–2 vs. daily: 4.507 (1.396–14.548) *
Ceber et al. (2005) [21]	Turkey	Case control	243 (123/120)	NR	Vegetable oil (margarine) Rarely vs. almost daily: 4.92 (1.92–12.59) * Olive oil Rarely vs. almost daily: 0.34 (0.16–0.73) * Animal fat: NS

OR (95% CI): odds ratio (confidence interval), significance threshold *p* < 0.05: *, these results were statistically significant, NS: no significant association; NR: not reported; reference: ref., t/w: times/week, g: gram, mg: milligram, MUFAs: monounsaturated fatty acids.

### 3.9. Carbohydrate Intake

Table 9 provides an overview of three studies in the MENA region that examined the relationship between carbohydrate intake and BC risk, with most research indicating no significant association. Alboghobeish et al. investigated the impact of carbohydrate intake on BC risk among 408 Iranian women (136 case) and found no significant association [34]. Similarly, Hosseini et al., in a study involving 300 participants (150 BC cases), and Sasanfar et al., in a study with 956 participants (461 BC cases), both reported that neither the quality nor the quantity of dietary carbohydrates had a significant influence on BC risk [35,36].

**Table 9 nutrients-17-00394-t009:** Findings of the included studies about the association between carbohydrate Intake and the risk of BC.

Reference	Country	Study Design	Sample Size (Cases/Controls)	Adjustment for Confounding Variables	Results
Alboghobeish et al. (2020) [34]	Iran	Case control	408 (136/272)	M1: Age. M2: Age, age at first pregnancy, BMI, family history of BC, physical activity, and total energy intake. M 3: Age at first pregnancy, BMI, family history of BC, Physical activity, total energy intake, total fiber intake, smoking, education level, and menopausal status.	Carbohydrate Overall: NS. Premenopausal: NS Postmenopausal: NS
Hosseini et al. (2021) [35]	Iran	Case control	300 (150/150)	M1: Age and energy intake. M2: Physical activity, education, marital status, socioeconomic status, alcohol use, smoking, vitamin supplements, medication use, comorbidities, length of oral contraceptives use, hormone replacement therapy, systolic BP, diastolic BP, age at menarche, time since menopause in post-menopausal women, weight at age 18 years old, number of children, breast feeding ages, and family history of BC. M3: Vitamin E, iron, vitamin B6, folic acid, and vitamin A. M4: BMI.	T3 vs. T4 Total carbohydrate: NS
Sasanfar et al. (2021) [36]	Iran	Case control	956 (461/495)	Age, energy, and residential place	Q1 vs. Q4 Carbohydrate intake Overall: NS Premenopausal: NS Postmenopausal: NS

NS: no significant association.

### 3.10. Vitamins Intake

A total of three studies examined the relationship between various vitamins and BC risk in the MENA region as detailed in Table 10. A study conducted by Alim et al. in Turkey assessed different dietary components and their association with BC risk in a small simple size 40 BC case and 40 controls, and suggested a protective effect of vitamin C, while other nutrients, such as vitamins A and E, showed no significant associations [33]. A study of 176 Iranian women (60 BC cases) conducted by Bidgoli et al. and reported that vitamin D supplements were associated with a slightly increased risk of BC, whereas vitamin D intake from eggs appeared to be protective, and vitamin D from fish showed no significant association [37]. However, another study in Iran conducted by Jamshidinaeini et al. among 270 women (135 BC case) found that dietary vitamin D had an overall protective effect against BC, while no significant associations were observed when analyzing premenopausal and postmenopausal women separately [38].

**Table 10 nutrients-17-00394-t010:** Findings of studies assessed the association between vitamins and minerals intake and the risk of BC in the MENA region.

Reference	Country	Study Design	Sample Size (Cases/Controls)	Adjustment for Confounding Variables	Results OR (95% CI)
Ebrahimpour-koujan et al. (2021) [32]	Iran	Case control	1050 (350/700)	M1: Age and energy. M2: Marital status, socio-economic status, education, family history of BC, menopausal status, breast-feeding, alcohol, smoking, and physical activity. M3: Saturated fat, trans fat, fiber, vitamin C and supplement use. M4: Body mass index. M5: Age, marital status, SES, education, family history of BC, breast-feeding, alcohol, smoking, and physical activity.	Q1 vs. Q4 Calcium intake Overall: NS Premenopausal: NS Postmenopausal: NS
Alim et al. (2016) [33]	Turkey	Case control	80 (40/40)	Menarche age, age at first birth, number of children, menopause age, energy, and BMI.	Vitamin A: NS vitamin E: NS vitamin C: 0.970 (0.956–0.983) *
Al Qadire et al. (2018) [12]	Jordan	Case control	823 (405/418)	Physical activity, calcium intake, and breast cancer self-examination, BMI, total caloric intake, employment, income, education level, family size, smoking, physical activity, cancer background, menstruation start, and contraceptives usage.	Calcium supplement intake: 2.15 (1.45 to 3.17) *
Bidgoli et al. (2014) [37]	Iran	Case control	176 (60/116)	Age at marriage, height at 18 yrs, age at highest weight, age at first pregnancy, menstrual Disorders	Calcium Supplements: 0.07 (0.01–0.58) * Vit D Supplements: 1.115 (1.049–1.187) * Vit D from fish: NS vit D from egg: 0.232 (0.067–0.806) *
Jamshidinaeini et al. (2016) [38]	Iran	Case control	270 (135/135)	M1: Calories, fat, calcium intake, age, body mass index, menopausal status, education, use of exogenous hormones, and duration of sun exposure.	Q1 vs. Q4 Total Vitamin D Overall: NS Pre-menopausal: NS Post-menopausal: NS Dietary Vitamin D Overall Crude: 0.39 (0.196–0.784) * M1: 0.38 (0.181–0.827) * Pre-menopausal: NS Post-menopausal: NS

OR (95% CI): odds ratio (confidence interval), significance threshold *p* < 0.05: *, these results were statistically significant, NS: no significant association.

### 3.11. Calcium Intake

As described in Table 10, three studies investigated the relationship between calcium intake and BC risk in the MENA region and have reported inconsistent results. One study conducted among 823 Jordanian women (405 BC case) found a positive association, suggesting that higher calcium supplements intake may be linked to an increased risk of BC [12]. Conversely, a case-control study from Iran involving 176 participants (60 BC cases) indicated a negative association between higher calcium supplement intake and breast cancer risk [37]. However, another study involving 1050 Iranian women (350 BC cases) found no significant association between dietary calcium intake and BC risk [32].

### 3.12. Dietary Patterns

In the MENA region, several studies have investigated the association between dietary patterns, dietary scores, and indices with BC risk, yielding mixed results. Three studies assessed the relationship between healthy and unhealthy dietary patterns and BC risk, with generally consistent findings as summarized in Table 11. Both Karimi et al., in a study with 274 participants (100 BC cases), and Heidari et al., in a study with 401 participants (134 BC cases), reported significant inverse associations between adherence to healthy dietary patterns and BC risk. In contrast, unhealthy dietary patterns were linked to an increased risk of BC in their studies. However, Tiznobeyk et al., in a study involving 150 participants (80 BC cases), observed a similar protective effect of healthy dietary patterns but found no significant association between unhealthy dietary patterns and BC risk [39,40,41].

**Table 11 nutrients-17-00394-t011:** Findings of studies assessing the association between dietary patterns identified through posteriori methods and the risk of BC in the MENA region.

Reference	Country	Study Design	Sample Size (Cases/Controls)	Adjustment for Confounding Variables	Results OR (95% CI)
Healthy Dietary Pattern					
Tiznobeyk et al. (2016) [39]	Iran	Case control	150 (80/70)	M1: Age (years), BMI (kg/m^2^), and energy intake (kcal). M2: Was adjusted for age at first pregnancy, job status and history of oestrogen therapy. M3: Was adjusted for the variables included in the models 1 and 2.	T1 vs. T3 Healthy dietary pattern Crude: 0.44 (0.20, 0.95) * M3: NS
Karimi et al. (2014) [40]	Iran	Case control	274 (100/174)	M1: Age and menopausal status. M2: Age at menarche, age at first full-term pregnancy, smoking status, oral contraceptive drug use, BMI, physical activity, family history of BC and relative accuracy of energy reporting. M3: Age, age at menarche, age at first full-term pregnancy, smoking status, oral contraceptive drug use, BMI, energy intake, physical activity, and family history of BC.	T1 vs. T3 Healthy dietary pattern Overall M1: NS M2: 0.25 (0.08, 0.78) * Premenopausal M3: 0.21 (0.4, 1.17) * Postmenopausal M3: 0.13 (0.02, 1.00) *
Heidari et al. (2018) [41]	Iran	Case control	401 (134/267)	M1: Age. M2: Age, age at first live birth, day bra uses, and vitamin D supplements. M3: Age, BMI, energy intake, MET, age at first live birth, day bra use, vitamin D supplements, and family history of cancer.	Q1 vs. Q4 Healthy dietary pattern Overall M1: 0.57 (0.32–1.01) * M2: 0.56 (0.30–1.04) * M3: NS Premenopausal: NS Postmenopausal: NS
Unhealthy Dietary Patterns					
Tiznobeyk et al. (2016) [39]	Iran	Case control	150 (80/70)	M1: Age, BMI, and energy intake (kcal). M2: Was adjusted for age at first pregnancy, job status, and history of oestrogen therapy. M3: Was adjusted for the variables included in the models 1 and 2.	T1 vs. T3 Unhealthy dietary pattern: NS
Karimi et al. (2011) [40]	Iran	Case control	274 (100/174)	M1: Age and menopausal status. M2: Age at menarche, age at first full-term pregnancy, smoking status, oral contraceptive drug use, BMI, physical activity, family history of BC and relative accuracy of energy reporting. M3: age, age at menarche, age at first full-term pregnancy, smoking status, oral contraceptive drug use, BMI, energy intake, physical activity, and family history of BC.	T1 vs. T3 Unhealthy dietary pattern M1: 5.94 (2.74, 12.89) * M2: 7.78 (2.31, 26.22) * Premenopausal M3: 18.82 (2.06, 171.6) * Postmenopausal M3: 42.07 (3.90, 454.2)
Heidari et al. (2018) [41]	Iran	Case control	401 (134/267)	M1: Age. M2: Age, age at first live birth, day bra uses, and vitamin D supplements. M3: Age, BMI, energy intake, MET, age at first live birth, day bra use, vitamin D supplements, and family history of cancer.	Q1 vs. Q4 Unhealthy dietary pattern Overall M1: NS M3: 2.21 (1.04–4.69) * Premenopausal: NS Postmenopausal M3: 3.56 (1.16–10.95) *
Nutrient patterns					
Tayyem et al. (2019) [42]	Jordan	Case control	400 (200/200)	Age, marital status, education, work, income, physical activity, smoking, family history, health problem, number of pregnancies, lactation, contraceptives, and hormonal replacement therapy.	Q1 vs. Q4 1st nutrient pattern 5.4 (2.11, 13.91) * 2nd nutrient pattern 0.67 (0.28, 1.64) * 3rd nutrient pattern 3.87 (1.53, 9.77) *
Fereidani et al. (2019) [43]	Iran	Case control	401 (134/267)	M1: age M2: age, height, age of first pregnancy, cancer family history and vitamin D supplement	1st nutrient pattern M1: 0.51 (0.33–0.80) * M2: 0.52 (0.32–0.82) * 2nd nutrient pattern M1: 0.84 (0.55–1.28) * M2: 0.81 (0.52–1.27) * 3th nutrient pattern M1: 0.64 (0.42–0.98) * M2: 0.66 (0.42–1.04) * 4th nutrient pattern M1: 1.22 (0.80–1.86) * M2: 0.13 (0.72–1.77) *

OR (95% CI): odds ratio (confidence interval), significance threshold *p* < 0.05: *, these results were statistically significant, NS: no significant association.

Table 12 below summarizes the findings of the included studies examining the association between dietary patterns identified using a priori methods and BC risk in the MENA region. The sample sizes of these studies ranged from 275 to 1050 participants. The relationship between the MedDiet and BC risk has been investigated in two studies in the MENA region. Firstly, a study conducted among a 1050 women (350 BC case) by Sadeghi et al. [44] and, secondly, a study by Djafari et al. involving 300 women (150 BC case) [45], where both reported a significant protective association. On the other hand, research on the Mediterranean Intervention for Neurodegenerative Delay (MIND) diet, which combines elements of the Mediterranean and the dietary approaches to stop hypertension (DASH) diet, shows conflicting results. Aghamohammadi et al., in a study involving 1050 women (350 BC cases) [46] suggested significant protective effects, while Sheikhhossein et al. in a study of 300 participants (150 BC cases) reported no significant association [47]. The DASH diet has shown consistent protective effects against BC, as demonstrated in two studies by Soltani et al. [48] involving 1050 participants (350 BC cases) and Heidari et al. [49] involving 401 participants (134 BC cases).

The dietary diabetes risk reduction score (DDRRS) has also been studied in the MENA region. Ebrahimi Mousavi et al., in a study involving 1050 participants (350 BC cases), observed significant protective effects against BC [50], while, Mohammadzadeh et al., in a study of 408 participants (136 BC cases), reported that the protective effect of DDRRS was significant only in postmenopausal women, with no significant association found in overall or premenopausal women [51].

In examining the association between plant-based diet index (PDI) and BC risk within the MENA region, three studies from Iran provide valuable insights, though with varying results. Payandeh et al. in study involving 300 participants (150 BC), found no significant association between the PDI, healthy plant-based diet index (hPDI), or unhealthy plant-based diet index (uPDI) and BC [52]. Conversely, a study conducted among 868 Iranian women (412 BC case) by Sasanfar et al. reported a significant inverse association between adherence to a hPDI and BC risk, whereas no significant associations were found for PDI and uPDI [53]. Additionally, in a study involving 1050 participants (350 BC cases), Rigi et al. [54] reported a strong inverse association between higher adherence to both the PDI and hPDI and a lower BC risk, specifically in both overall participants and post-menopausal women. They also found a significant positive association between higher adherence to the uPDI and an increased BC risk.

Dietary insulin index and load, which measure the insulin response to food, have been linked to BC risk in the MENA region studies. Notably, Akbari et al. in a study involving 500 participants (250 BC cases) [55], and Sheikhhossein et al. in a study involving 300 participants (150 BC cases) [56], found significant positive associations, indicating higher risk.

Studies assessing the impact of carbohydrate quality and quantity found no significant associations between the low-carbohydrate diet score and BC risk, as reported by Hosseini et al. [35] and Sasanfar et al., [57] in their studies involving 300 participants (150 BC cases) and 1009 participants (486 BC cases), respectively. However, the carbohydrate quality index was associated with a reduced risk of BC in the study by Hosseini et al.

Safabakhsh et al. conducted a study in Iran among 300 women (150 BC cases) and found no significant association between dietary total antioxidant capacity (TAC) and BC risk overall in either pre-menopausal or post-menopausal women [58]. Similarly, Karimi et al., in study involving 275 women (100 BC case), reported no significant overall association between TAC and BC. Nonetheless, they observed a protective effect of TAC from fruit [59]. Sasanfar et al. also noted no overall significant association for TAC but identified a significant protective effect in postmenopausal women [60]. Jalali et al., in a study involving 308 participants (136 BC), found a non-significant association between TAC using Ferric Reducing Antioxidant Power (FRAP) and BC risk across all groups, though TAC using the Antioxidant Capacity (AC) method showed a crude significant association that was not maintained after adjustment [61].

In addition, in this systematic review, several studies investigated the association of dietary inflammatory index (DII) with BC in the MENA region with a simple size ranging from 293 to 2011 participants. The DII consistently showed a positive association with BC risk, highlighting the potential role of inflammation in cancer development [62,63,64,65,66,67]. Conversely, only one study did not find significant associations between the DII and BC risk across [68].

**Table 12 nutrients-17-00394-t012:** Findings of studies assessing the association between dietary patterns and indices identified through a priori methods with the risk of BC in the MENA region.

Reference	Country	Study Design	Sample Size (Cases/Controls)	Adjustment for Confounding Variables	Results OR (95% CI)
MedDiet					
Sadeghi et al. (2023) [44]	Iran	Case control	1050 (350/700)	M1: Age and energy. M2: Additionally, adjusted for region, marital status, education, disease history, physical activity, family history of BC, menopausal status, smoking, alcohol consumption, and socioeconomic status. M3: Additional adjustment for BMI.	T3 vs. T1 MedDiet scores Overall Crude: 0.34 (0.23–0.48) * M3: 0.43 (0.28–0.67) * pre-menopause: NS post-menopause Crude: 0.33 (0.22–0.49) M3: 0.37 (0.23–0.60) *
Djafari et al. (2023) [45]	Iran	Case control	300 (150/150)	M1: Age and energy intake M2: Education, residency, family history of BC, physical activity, marital status, smoking, alcohol consumption, supplement use, length of breast-feeding, menopausal status, and history of hormone replacement therapy. M3: BMI.	Q1 vs. Q4 MedDiet Quality Index Overall Crude: 0.47 (0.23, 0.92) * M1: 0.47 (0.24, 0.93).* M2: 0.45 (0.21, 1.94) * M3: 0.45 (0.21, 0.94) * Premenopausal: NS Postmenopausal Crude: NS M1: 0.24 (0.07, 0.8)) * M2: NS M3: NS
Western dietary pattern					
Foroozani et al. (2022) [69]	Iran	Case control	2018 (1009/1009)	M1: Energy intake, family history of BC, smoking status, OCP, chest X-ray, history of benign breast disease, BMI, physical activity, age at first delivery (year), breastfeeding (month), history of miscarriage, menarche age (year), and menopausal status. M2: Adjusted for M1 + fruit and vegetable intakes.	T3 vs. T1 Western dietary pattern. Invasive ductal carcinoma (IDC) All participants Crude: 0.79 (0.56, 1.03) * M2: 2.45 (1.88, 3.17) * pre-menopause Crude: NS M2: 2.95 (1.91, 4.56) * post-menopause Crude: NS M2: 2.16 (1.39, 3.37) * Invasive lobular carcinoma (ILC) All participants Crude: 1.22 (0.77, 1.66) * M2: NS Post-menopause Crude: 0.43 (0.02, 1.10) * M2: 1.35 (0.64, 2.85) * Pre-menopause Crude: 2.03 (1.39, 2.68) * M2: 5.25 (0.54, 10.82) *
MIND diet					
Sheikhhossein et al. (2020) [47]	Iran	Case control	300 (150/150)	M1: Age and energy intake. M2: Education, marital status, menopause status, age at first pregnancy, socioeconomic status, alcohol use, smoking, vitamin supplements and medicines uses, medical history, history of hormone and OCP use, age at first menarche, time since menopause in post-menopausal women, weight at age 18 years old, number of children, length of breast feeding, and family history of BC. M3: Vitamin E, iron, vitamin B6, folic acid, and vitamin A. M4: BMI.	T1 vs. T3 MIND diet score Overall: NS Premenopausal: NS Postmenopausal: NS Q1 vs. Q4 MIND diet score Overall Crude: NS M3: 0.50 (0.34–0.72) * Premenopausal: NS Postmenopausal Crude: 0.52 (0.37–0.71) * M3: 0.45 (0.30–0.66) *
Aghamohammadi et al. (2020) [46]	Iran	Case control	1050 (350/700)	M1: Age and energy intake. M2: Additionally, adjusted for education, SES, residency, family history of BC, physical activity, marital status, smoking status, alcohol consumption, supplement use, breast-feeding, and menopausal status. M3: Additionally adjusted for BMI.	
PDI					
Payandeh et al. (2021) [52]	Iran	Case control	300 (150/150)	M1: Age and energy intake and body mass index. M2: M1 additionally adjusted for physical activity, family history of cancer, socioeconomic status, smoking, alcohol consumption, menopause status, first menstruation age, weight at 18 years old, length of breastfeeding, hormone replacement therapy, dietary supplement use, medication use, and comorbidities.	T1 vs. T3 PDI Overall: NS Pre-menopause: NS Post-menopause: NS hPDI Overall: NS Pre-menopause: NS Post-menopause: NS uPDI Overall: NS Pre-menopausal: NS Post-menopausal: NS
Sasanfar et al. (2021) [53]	Iran	Case control	868 (412/456)	M1: Age and energy. M2: Adjusted for age, energy, physical activity, family history of BC, education, parity, and marital status. M3: Further adjusted for body mass index (BMI).	Q1 vs. Q4 PDI Overall: NS pre-menopause: NS post-menopause: NS hPDI Overall Crude: 0.63 (0.43–0.93 * M3: 0.61 (0.40–0.93) * pre-menopausal M1: 0.55 (0.37–0.83) * post-menopause M1: 0.56 (0.37–0.86) * M3: 0.63 (0.40–0.98) * uPDI Overall: NS pre-menopausal: NS Post-menopausal: NS
Rigi et al. (2021) [54]	Iran	Case control	1050 (350/700)	M1: Age and energy intake. M2: Additionally, adjusted for education, social economic status, residential area, supplement use, family history of BC, disease history, physical activity, marital status, smoking status, alcohol consumption, and history of breast-feeding and menopausal status. M3: Further adjustment for BMI.	Q4 vs. Q1 PDI Overall: Crude: 0.23 (0.15–0.33) * M3: 0.33 (0.22–0.50) * Pre-menopausal: Crude: 0.20 (0.07–0.56) M3: NS Post-menopausal: NS Crude: 0.24 (0.16–0.35) * M3: 0.35 (0.22–0.56) * hPDI Overall: Crude: 0.57 (0.40–0.81) * M3: 0.64 (0.43–0.94) * Pre-menopause: NS Post-menopause: Crude: 0.53 (0.36–0.78) * M3: 0.62 (0.41–0.95) * uPDI Overall: Crude: 2.12 (1.46–3.08) * M3: 2.23 (1.48–3.36) * Pre-menopause: NS Post-menopause: Crude: 2.42 (1.60–3.65) * M3: 2.42 (1.51–3.87) *
DASH					
Soltani et al. (2020) [48]	Iran	Case control	1050 (350/700)	M1: Age and energy intake. M2: Education, residency, family history of BC, physical activity, marital status, smoking, alcohol consumption, supplement use, breast-feeding, and menopausal status. M3: BMI.	Q1 vs. Q4 DASH diet score Overall Crude: 0.13 (0.08–0.20) * M1: 0.12 (0.08–0.19) * M2: 0.13 (0.08–0.20) * M3: 0.15 (0.09–0.24) * Premenopausal: NS Postmenopausal Crude: 0.09 (0.05–0.15) * M1: 0.09 (0.05–0.14) * M2: 0.09 (0.05–0.16) * M3: 0.11 (0.06–0.19) *
Heidari et al. (2020) [49]	Iran	Case control	401 (134/267)	Age, BMI, energy intake, physical activity, age at first live birth, vitamin D supplements, and family history of cancer.	Q1 vs. Q5 Dixon’s DASH index Overall: NS Premenopausal: NS postmenopausal: NS Mellen’s DASH index Overall: 0.50 (0.62–0.97) * Premenopausal: NS Postmenopausal: 0.24 (0.08–0.68) * Fung’s DASH index Overall: NS Premenopausal: NS Postmenopausal: 0.36 (0.13–0.94) * Günther’s DASH index Overall: 0.48 (0.25–0.93) * Premenopausal: NS Postmenopausal: NS
Dietary insulin index and load					
Akbari et al. (2021) [55]	Iran	Case control	500 (250/250)	M1: Age and BMI. M2: Waist, physical activity, smoking tobacco, and dietary intake of energy. M3: Adjusted for M2 and anti-inflammatory drugs use, family history of BC, family history of cancer, vitamin D supplement, herbal drug use, constant use of OCP, hormone therapy, menopausal status, age at menopause, benign breast diseases history, inflammatory disease history, night bra use, and comorbidity.	T1 vs. T3 Dietary Insulin Index Crude: 3.56 (1.85–6.85) * M3: 1.46 (0.67–319) * Dietary Insulin Load Crude: 2.65 (1.43–493) * M3: 1.87 (0.92–380) * T1 vs. T3 Dietary Insulin Index Crude: 1.82 (1.02–3.25) * M1: 1.86 (1.03–3.35) * M2: NS M3: NS M4: NS Dietary Insulin Load Crude: 1.9 (1.06–3.40) * M1: 2.07 (1.14–3.76) * M2: NS M3: NS M4: NS
Sheikhhossein et al. (2021) [56]	Iran	Case control	300 (150/150)	M1: Age and energy intake. M2: Education, marital status menopause status, socioeconomic status, alcohol use, smoking, use of vitamin supplements and medicines, medical history, history of hormone and oral contraceptive use, age at first menarche, time since menopause in postmenopausal women, weight at age 18 years, number of children, length of breast feeding, and family history of BC. M3: Vitamin E, iron, vitamin B6, folic acid, and vitamin A intake. M4: Body mass index.	
Glycemic Index and load					
Alboghobeish et al. (2020) [34]	Iran	Case control	408 (136/272)	M1: Age. M2: Age, age at first pregnancy, BMI, family history of BC, physical activity, total energy intake. M3: Age at first pregnancy, BMI, family history of BC, Physical activity, total energy intake, total fiber intake, smoking, education level, and menopausal status	Q1 vs. Q4 Glycemic Index Overall M3: 2.49 (1.28–4.82) * Premenopausal: NS Postmenopausal M3: 4.45 (1.59–12.47) * Glycemic Load Overall: NS Premenopausal: NS Postmenopausal: NS
Rigi et al. (2022) [70]	Iran	Case control	1050 (350/700)	M1: Education, socio-economic status, urban-resided, supplement use, family history of BC, physical activity, marital status, smoking status, alcohol consumption, breastfeeding, and menopausal status. M2: Age and energy intake. M3: Additional adjustment for BMI.	T1 vs. T3 Dietary Glycemic index Overall Crude: 1.40 (1.02, 1.91) * M3: 1.47 (1.02, 2.12) * Premenopausal: NS Postmenopausal Crude: 1.50 (1.06, 2.11) * M3: 1.51 (0.98, 2.06) * Dietary Glycemic load Overall: NS Premenopausal: NS Postmenopausal Crude: 1.48 (1.05, 2.08) * M3: NS
Hosseini et al. (2022) [35]	Iran	Case control	300 (150/150)	M1: Age and energy intake. M2: Physical activity, education, marital status, socioeconomic status, smoking, vitamin supplements and medication use, comorbidities, length of oral contraceptives use, hormone replacement therapy, age at menarche, time since menopause in post-menopausal women, weight at age 18 years old, number of children, breast feeding ages, and family history of BC. M3: Vitamin E, iron, vitamin B6, folic acid, and vitamin A. M4: BMI.	Glycemic index: NS Glycemic load: NS
Carbohydrates diet					
Hosseini et al. (2022) [35]	Iran	Case control	300 (150/150)	M1: Age and energy intake. M2: Physical activity, education, marital status, socioeconomic status, smoking, vitamin supplements and medication use, comorbidities, length of oral contraceptives use, hormone replacement therapy, systolic BP, diastolic BP, age at menarche, time since menopause in post-menopausal women, weight at age 18 years old, number of children, breast feeding ages, and family history of BC. M3: vitamin E, iron, vitamin B6, folic acid, and vitamin A. M4: BMI.	T1 vs. T4 LCDS Crude: 0.46 (0.263–0.815) * M1: NS M2: NS M3: NS M4; NS T1 vs. T4 CQI Crude: 0.46 (0.263–0.815) * M1: 0.46 (0.264–0.818) * M2: 0.48 (0.26–0.913) * M3: 0.42 (0.216–0.822) * M4: 0.41 (0.213–0.821) *
Sasanfar et al. (2019) [57]	Iran	Case control	1009 (486/523)	M1: Age. M2: Physical activity, family history of BC, menopausal hormone use, education, parity, oral contraceptive use, cigar smoking, alcohol consumption, fertility treatment, marital status M3: Vitamin E, iron, vitamin B6, folic acid, and vitamin A. M4: BMI.	Q1 vs. Q4 LCDS Overall: NS Premenopausal: NS Postmenopausal: NS
Dietary Diabetes Risk Reduction Score (DDRRS)					
Ebrahimi Mousavi et al. (2022) [50]	Iran	Case control	1050 (350/700)	M1: Unadjusted. M2: Age, residence, marital, menopausal and socioeconomic status, education, family history of BC, breast feeding, history of disease, and dietary supplement use. M3: BMI.	DRRDS M1: 0.66 (0.46, 0.95) * M2: 0.59 (0.39, 0.87) * M3: 0.59 (0.38, 0.90) * pre-menopause: NS post menopause M1: 0.63 (0.43, 0.93) * M2: 0.55 (0.36, 0.85) * M3: 0.57 (0.36, 0.90) *
Mohammadzadeh et al. (2023) [51]	Iran	Case control	408 (136/272)	_	DRRDS Overall: NS Premenopausal: NS Postmenopausal: 0.43 (0.19–0.99) *
Dietary Inflammatory index					
Sohouli et al. (2022) [62]	Iran	Case control	520 (253/267)	M1: Age and BMI. M2: Waist circumference, energy, age at first pregnancy, number of children, history of abortion, use of anti-inflammatory drugs, and vitamin supplements D. M3: Age and waist circumference M4: Energy, first pregnancy age, number of children, history of abortion, use of anti inflammatory drugs, and vitamin supplements D.	Q1 vs. Q4 DIS Crude: 2.56 (1.48–4.44) * M2: 2.13 (1.15–3.92) * EDII Crude: NS M4: 2.17 (1.12–4.22) *
Gholamalizadeh et al. (2022) [63]	Iran	Case control	540 (180/360)	M1: Age. M2: BMI, alcohol consumption, smoking, pregnancy number, abortion number, breastfeeding duration, menopause age, and total calorie intake	DII M1: 2.11 (1.01–4.46) * M2: 5.02 (1.43–17.58) *
Vahid et al. (2018) [64]	Iran	Case control	293 (145/148)	M1: Age and energy adjusted. M2: Age, energy, education, exercise, BMI, smoking, family history of cancer, age at menarche, parity, marital status, menopausal status, oral contraceptive use and hormone replacement therapy	T1 vs. T3 DII M1: 1.76 (1.43, 2.18) * M2: 1.80 (0.08–0.20) *
Hammad et al. (2021) [68]	Jordan	Case control	400 (200/200)	M1: Age. M2: Age, total energy. M3: Age, education, total energy, body mass index, number of pregnancies, contraceptive use, lactation, smoking, and family history of BC.	T1 vs. T3 DII: NS
Jalali et al. (2018) [65]	Iran	Case control	408 (136/272)	M1: Age. M2: Age and energy. M3: Age, age at first pregnancy, height, family history of cancer, day bra wearing, menopausal status, nonsteroidal anti-inflammatory. drugs, and vitamin D supplement. M4: Adjusted for1 M3 + energy. M5: Adjusted for M4 + breastfeeding time, BC family history, night bra wearing, smoking status, BMI, educational level, marital status, hormone replacement therapy (HRT), and physical activity.	Q1 vs. Q4 DII Crude: 2.7 (1.49–4.98) * M5: 2.6 (1.12–6.25) *
Ghanbari et al. (2022) [66]	Iran	Case control	300 (150/150)	M1: Age and energy. M2: Education, Marital status, Occupation Physical activity, alcohol, smoking, Vitamin supplement, History o hormonal use, first menstruation age, number of children, menopause status, and medical history. M3: Additional adjustment for BMI.	Q1 vs. Q4 FDII Crude: 2.38 (1.23–4.59) * M2: 2.8 (1.20–4.55) *
Hayati et al. (2022) [67]	Iran	Case control	2011 (1007/1004)	M1: Menopausal status, age at first pregnancy type 2 diabetes, BMI, total duration of breastfeeding, average duration of lactation for each child, the number of breastfed children, abortion history, and total energy intake. M2: Menopausal status, age at first pregnancy, type 2 diabetes, BMI, total duration of breastfeeding, average duration of each lactation, and breastfeeding history. M3: The adjustment was performed according to M2, with the replacement of 2 confounders (average duration of lactation for each child and breast-feeding history) by abortion history and total energy intake. M4: The adjustment was performed according to M2 with the exclusion of breastfeeding history.	Q1 vs. Q4 DII Crude: NS M1: 1.56 (1.11–2.18) * Energy-adjusted DII Crude: 1.64 (1.28–2.09) * M2: 1.87 (1.42–2.47) * Energy-adjusted DII including supplements: Crude: 1.57 (1.22–2.00) * M4: 1.94 (1.42–2.65) *
Dietary Antioxidant Index					
Safabakhsh et al. (2020) [58]	Iran	Case control	300 (150/150)	Age, BMI, physical activity, education, marital status, socioeconomic status, alcohol use, smoking, vitamin supplements and medication use, comorbidities, length of oral contraceptives (OCP) use, hormone replacement therapy, systolic BP, diastolic BP, age at menarche, time since menopause in post-menopausal women, weight at age 18 years old, number of children, breast feeding ages, family history of BC, dietary intake of fibre, tea, coffee and total energy.	TT1 vs. T3 TAC Overall: NS Pre-menopausal: NS Post-menopausal: NS
Karimi et al. (2015) [59]	Iran	Case control	275 (100/175)	M1: Age. M2: Age at menarche, age at first pregnancy, number of full pregnancies, smoking, use of oral contraceptives and use of brassiere per day. M3: Body mass index and life satisfaction. M4: Menopause status, family history of BC, physical activity, energy intake, and energy density of the diet,	Q1 vs. Q4 TAC: NS TAC of Fruit (g/day) M1: NS M4: 0.29 (0.13–0.68) * TAC of Legumes (g/day): NS
Sasanfar et al. (2020) [60]	Iran	Case control	1030 (503/506)	age, body mass index, menopausal status. M1: Age and energy. M2: Physical activity, family history of BC, menopausal hormone use, education, parity, oral contraceptive use, cigar smoking, alcohol consumption, fertility treatment, marital status, folic acid, B6 and BMI.	Q1 vs. Q4 TAC Overall: NS Premenopausal: NS Postmenopausal M1: 0.47 (0.24, 0.93) * M2: 0.28 (0.11, 0.72) *
Jalali et al. (2022) [61]	Iran	Case control	308 (136/272)	M1: Age, first pregnancy age, menopausal status. M2: Adjusted for model 1 in addition to energy.	Q1 vs. Q4 TAC by FRAP Overall: NS Premenopausal: NS Postmenopausal: NS TAC by AC Overall Crude: 0.52 (0.28–0.97) * M2: NS Premenopausal: NS Postmenopausal: NS
Allahyari et al. (2022) [71]	Iran	Case control	540 (180/360)	M1: Age M2: BMI, the number of pregnancies, duration of breastfeeding, menopause age, and total energy intake	DAI M1: NS M2: 0.91 (0.90–0.93) *
Vahid et al. (2022) [72]	Iran	Case control	293 (145/148)	M1: Age. M2: Age, education, BMI, occupation, alcohol consumption, smoking, pregnancy, family history, menarche age, MET, HRT, and total calorie intake	DAI < −0.6 vs. DAI ≥ −0.6 DAI Crude: 0.55 (0.34–0.88) * M2: 0.18 (0.09–0.37) *
HEI					
Ebrahimpour-Koujan et al. (2024) [73]	Iran	Case control	1050 (350/700)	M1: Age and energy. M2: Additional adjustment for residence, marital status, SES, education, family history of B.C, history of disease, menopausal status, breast feeding, smoking, physical activity and supplement use. M3: Further adjustment for BMI	Q1 vs. Q4 HEI score Overall Crude: 0.40 (0.27–0.57) * M1: 0.22 (0.14–0.33) * M3: 0.27 (0.17–0.43) * Premenopausal: Crude: NS M1: 0.23 (0.08–0.67) * M3: NS Postmenopausal: Crude: 0.39 (0.26–0.56) * M1: 0.20 (0.13–0.32) * M3: 0.22 (0.13–0.37) *

OR (95% CI): odds ratio (confidence interval), significance threshold *p* < 0.05: *, these results were statistically significant, NS: no significant association, MedDiet: Mediterranean diet, MIND diet: Mediterranean intervention for neurodegenerative delay, PDI: plant-based diet index, uPDI: unhealthy plant-based diet index, hPDI: healthy plant-based diet index, DASH: dietary approaches to stop hypertension, LCDS: low carbohydrates diet score, CQI: carbohydrate quality score, DRRDS: diabetes risk reduction score, DIS: dietary inflammation score, DII: dietary inflammatory index, EDII: empirical dietary inflammatory index, FDII: food-based empirical dietary inflammatory index, TAC: total antioxidant capacity, DAI: dietary antioxidant index, HEI: healthy eating index.

## 4. Discussion

This systematic review primarily focused on dietary factors associated with BC risk within the MENA population. Although, it is essential to recognize that BC is a multifactorial and complex disease, and that many risk factors could be involved in its development, including genetic predisposition, hormonal influences, environmental exposures, and lifestyle factors such as diet [74,75]. Nonetheless, understanding how dietary habits as a modifiable lifestyle factor influence BC risk within each population is essential for addressing effective prevention strategies and public health initiatives.

The MENA region has experienced a dietary shift from traditional diets rich in fruit vegetables, and whole grains to more Westernized diets high in processed foods, red meat, unhealthy fats, and refined carbohydrates during the past decades [6,7]. The present systematic review aimed to assess the current state of evidence for the relationship between various dietary factors (foods, nutrients, and dietary patterns) and BC risk among women in the MENA region. In total, 65 articles were included in this review, reporting findings of the association between different dietary components and patterns, and BC risk among women in the MENA region.

The findings from our systematic review strongly suggest that a high intake of fruit and vegetables may play a protective role against BC among women in the MENA region. The majority of the studies reviewed consistently indicated an inverse association between the consumption of fruit and vegetables and the risk of developing BC. This is supported by the nutrients content of fruit and vegetables, such as antioxidants (vitamin C, vitamin E, selenium, and beta-carotene), fiber, and phytochemicals. These components are known to contribute to cancer prevention by reducing oxidative stress and inflammation [76]. Additionally, previous evidence, including other systematic reviews, further supports the protective effect highlighted in our findings [77,78,79].

A protective effect of fish and seafood consumption has been observed in many studies within the MENA region, where traditional diets often include these foods. The decreased risk of BC is potentially attributed to omega-3 fatty acids in fish and seafood, which possess anti-inflammatory properties [80]. Additionally, fish and seafood are rich in antioxidants, which may neutralize free radicals and reduce oxidative stress, both involved in the development and progression of cancer [81]. Furthermore, in the context of the MENA region, the frequent inclusion of locally fresh fish and seafood in the diet may enhance the protective effects compared to regions where processed or frozen alternatives are more commonly consumed. Additionally, cultural dietary practices, such as cooking methods that preserve nutrient content, may amplify the benefits of these foods. Furthermore, similar findings from previous reviews in other populations also support the protective effect of fish and seafood in reducing BC risk suggested in this review [82,83].

Similarly, black tea, a beverage widely consumed in the MENA region, has been reported in studies conducted in this region to be associated with a reduced risk of BC. The frequent intake and cultural importance of black tea in the MENA region may contribute to a stronger exposure to its bioactive compounds such as catechins, theaflavins, and thearubigins that may neutralize free radicals, thereby reducing oxidative stress and subsequent DNA damage that can lead to cancer. These polyphenols have been shown to inhibit the growth of cancer cells and induce apoptosis in tumor cells, potentially preventing the initiation and progression of cancer [84,85]. Additionally, some studies suggest that these polyphenols can modulate estrogen metabolism by influencing the enzymes involved in the production of estrogen. This modulation can lead to a reduction in the levels of biologically active estrogen, thereby potentially decreasing the risk of hormone receptor-positive BC [86]. For broader context, findings from this review align with some of the observations reported in previous systematic reviews and meta-analyses, such as the one conducted by Sun et al., Their analysis highlighted variability in the association between tea consumption and BC risk. Notably, eight case-control studies, similar to those conducted in the MENA region, reported a reduced BC risk associated with black tea consumption, while five cohort studies indicated a modest increase in risk. However, these cohort studies were conducted in countries like the Netherlands, Japan, Sweden, and the United States, which differ significantly from MENA countries in factors such as genetic predispositions, lifestyle, and dietary habits [87]. For instance, black tea consumption is relatively lower in these countries compared to the MENA region, where it is a traditional and widely consumed daily beverage, particularly black tea [88]. Nonetheless, cohort studies in the MENA population are needed to confirm the findings of case-control studies and provide a more comprehensive understanding of this association.

Regarding the association between BC risk and meat consumption studies in the MENA region revealed inconsistent findings. Half of the included studies reported a positive association, while the other half found no significant link. However, a recent meta-analysis found that women who consumed red meat had an increased risk of BC compared to those who did not consume red meat [89]. The association between red meat consumption and BC risk may be due to the content of heme iron, which is found to promote oxidative stress and has been associated with cancer risk [90]. Another systematic review and meta-analysis of prospective studies reported that consumption of processed meat was associated with a 9% higher risk of BC [91]. It is noteworthy that cooking meat to a well-done level, regardless of the type of meat, causes the formation of heterocyclic amines and polycyclic aromatic hydrocarbons, which are known to be carcinogenic [92]. Regarding poultry consumption in the MENA region, the findings of the reviewed studies were inconsistent. Only one study suggested an increased risk of BC associated with poultry consumption, while two others reported no significant association. Similarly, evidence in the literature regarding the association between poultry consumption and BC risk remains limited and inconclusive. Some studies conducted outside the MENA region have found no significant association between poultry intake and BC, concluding that poultry may have a neutral effect on BC risk compared to red or processed meats, which are more strongly linked to cancer development [93,94]. In contrast, other research has suggested that poultry consumption could have a potential protective effect, attributed to its lower fat content and lack of harmful compounds such as heme iron, which is abundant in red meat and has been associated with oxidative stress and cancer risk [95]. In general, results of the included studies about meat consumption might suggest that red and processed meat intake could be associated with increased risk of BC in the MENA region, but further research is needed in this area to confirm this relationship.

Similarly to results of the present systematic review, evidence from the literature identified milk consumption as being associated with an increased risk of BC [96,97]. This could be due to various factors, such as the presence of hormones in milk, including estrogens and growth factors, which have been suggested to influence cancer risk [97,98]. However, other studies indicate that milk consumption has a neutral or even protective effect against BC [99], possibly due to the beneficial nutrients such as calcium and vitamin D found in it [100]. The relationship between milk consumption and BC risk remains complex and may be influenced by factors such as the type of milk (whole, low-fat, or skim), the amount consumed, and individual genetic and lifestyle factors [101].

White bread consumption was also reported in studies from MENA region, as a risk factor for BC. White bread is typically made from refined grains, which have been stripped from beneficial nutrients like fiber, vitamins, minerals, and phytochemicals during processing [102]. This results in a high glycemic index, meaning it can cause rapid spikes in blood sugar levels [103]. Some research suggests that diets high in refined carbohydrates and foods with a high glycemic index may be associated with an increased risk of cancer including BC [104]. This may be due to its high glycemic index, leading to increased blood sugar and insulin levels, which have been implicated in BC development [105]. However, the evidence is not entirely consistent [106], and other factors such as overall diet quality, lifestyle, and genetic predisposition also play significant roles in BC risk [75].

The MedDiet has been associated with a reduced risk of BC in the present systematic review, consistently aligning with several other studies [107,108]. The MedDiet’s rich content of antioxidants, fiber, and healthy fats, particularly omega-3 fatty acids from fish and monounsaturated fats from olive oil, contribute to its protective role against the development of BC. Additionally, the anti-inflammatory and hormone-regulating properties of the MedDiet are believed to further contribute to its potential in lowering BC risk [75,109].

Similarly, studies from the MENA region suggest that adherence to a hPDI is associated with a reduced risk of BC. Whereas findings for the overall PDI and uPDI have been inconsistent. PDI generally refer to a dietary pattern derived from plant sources, such as fruit, vegetables, nuts, seeds, and legumes, while minimizing or excluding animal products. However, not all plant-based diets provide the same health benefits, as their nutritional quality can vary widely based on the composition and processing of the plant-based foods included. This highlights the importance of not considering the overall PDI in studies focused on health outcomes. hPDI are characterized by a high consumption of minimally processed, nutrient-rich plant foods, and these diets provide a range of protective nutrients, such as dietary fiber, antioxidants, vitamins, minerals, polyphenols, and flavonoids [110]. Additionally, such diets are typically low in saturated fats and processed foods, which have been associated with an increased risk of BC [103].

A higher dietary antioxidant index, reflecting greater intake of antioxidant-rich foods, was related to a reduced risk of BC in the MENA region. Antioxidants help to neutralize free radicals, thereby reducing oxidative stress and potential DNA damage [111]. In line with findings of this systematic review, several studies suggest that a high DAI is associated with a lower risk of BC, as antioxidants may help prevent DNA damage and inhibit cancer cell proliferation [111,112]. However, the evidence is not entirely consistent, with some studies finding no significant association [113]. The relationship between DAI and BC risk is complex and may be influenced by various factors, including the types and sources of antioxidants, individual genetic predispositions, and lifestyle factors.

Conversely, a higher dietary insulin index and load, as well as the dietary glycemic index, were reported in the majority of studies as being associated with an increased risk of BC. The dietary insulin index measures how much certain foods raise insulin levels, while the insulin load takes into account the quantity of food consumed [103,114]. The glycemic index measures how quickly carbohydrates in food raise blood glucose levels [115]. Diets with a high insulin index, load, and glycemic index are thought to increase BC risk by promoting hyperinsulinemia and insulin resistance, conditions linked to cancer development [116]. Elevated insulin levels can enhance cell proliferation and inhibit apoptosis, contributing to cancer progression [114]. However, findings are not entirely consistent across all populations [117], and further research is needed to better understand these relationships and to account for confounding factors such as body mass index, physical activity, and overall dietary patterns.

DII measures the inflammatory potential of an individual’s diet based on the consumption of various food components. A higher DII index has been associated with an increased risk of BC in the present systematic review and other ones [118,119]. Pro-inflammatory diets typically include higher intakes of processed foods, red and processed meats, refined sugars, and saturated fats, while anti-inflammatory diets are rich in fruit, vegetables, whole grains, and healthy fats like those found in nuts and fish [118]. Chronic inflammation is known to play a role in the development and progression of cancer, including BC [120]. Therefore, adhering to a diet with a lower DII score, which highlights anti-inflammatory foods, may help reduce the risk of BC [121].

Overall, our findings were in line with other studies that also reported inverse associations of BC risk for foods with health enhancing properties such as fruit and vegetables, while unhealthy dietary patterns were associated with an increased risk of BC in the MENA region. These unhealthy dietary patterns often include high intakes of processed foods, sugars, refined grains, and unhealthy fats, which can contribute to obesity, inflammation, and other risk factors for cancer, all of which are risk factors for BC. Additionally, these dietary patterns often lack essential nutrients and antioxidants found in fruit, vegetables, whole grains, and healthy fats that have protective effects against cancer. The consumption of high-fat and high-sugar foods can lead to increased levels of estrogen and other hormones that may promote the development of BC.

Additionally, in this review, the evidence for certain food groups and dietary components, such as poultry, yogurt, cheese, and other dairy products, as well as cereals, carbohydrate intake, fat intake, and specific diets like the Western diet and DASH diet, was inconsistent or limited. This underscores the need for further research to clarify these associations.

This systematic review has several notable strengths. It provides a comprehensive synthesis of available data on dietary factors and BC risk amongst women in the MENA region, offering a region-specific perspective that is often underrepresented in global studies. By highlighting the diverse dietary habits influenced by cultural, socio-economic, and geographical factors, this review sheds light on the unique dietary patterns that may contribute to BC risk in the MENA populations. Additionally, the review’s rigorous methodology, including the thorough search and inclusion criteria, ensures that a wide range of relevant studies were considered. The emphasis on identifying gaps in the current research also provides a valuable direction for future studies, aiming to fill these gaps and improve our understanding of BC risk in this region. Nevertheless, a few limitations must be considered when interpreting the findings of this systematic review. The associations of BC with dietary factors are complex, especially considering that BC is a multifactorial disease with multiple risk factors involving genetics, as well as individual and environmental factors. Additionally, the long-term development of BC adds another layer of difficulty in establishing causality, as dietary habits and other exposures can change over time, complicating retrospective assessments. Moreover, the MENA region is characterized by diverse dietary habits influenced by cultural, socio-economic, and geographical factors, with significant differences in food consumption between various areas. This dietary diversity complicates the task of isolating specific dietary factors and their association with BC risk. Furthermore, the included studies utilized a range of dietary assessment methods, each with varying levels of details and different numbers of food items. Additionally, all the included studies were case control design, which are subject to limitations, due to its observational methodology, such as potential recall and selection bias. Moreover, the adjustment for potential confounders varied significantly between studies. This variability introduces challenges in standardizing and comparing findings across studies. These inconsistencies highlight a methodological limitation that may affect the reliability and comparability of the reported associations. Over and above that, there is a paucity of studies investigating the association between diet and BC across the entire MENA region, with most research being conducted in Iran, highlighting gaps in other MENA countries. It is right to recognise that, despite our efforts to include all relevant reports on BC in MENA, the potential for selection bias could not be neglected.

## 5. Conclusions

Overall, this systematic review highlights the complex and multifactorial nature of the relationship between dietary factors and BC risk in the MENA region. The insufficient and inconsistent findings across studies underscore the need for further high-quality, well-designed research to elucidate the specific dietary patterns and components that may influence BC risk in this population. Addressing these knowledge gaps can inform targeted dietary recommendations for BC prevention in the MENA region.

## Figures and Tables

**Figure 1 nutrients-17-00394-f001:**
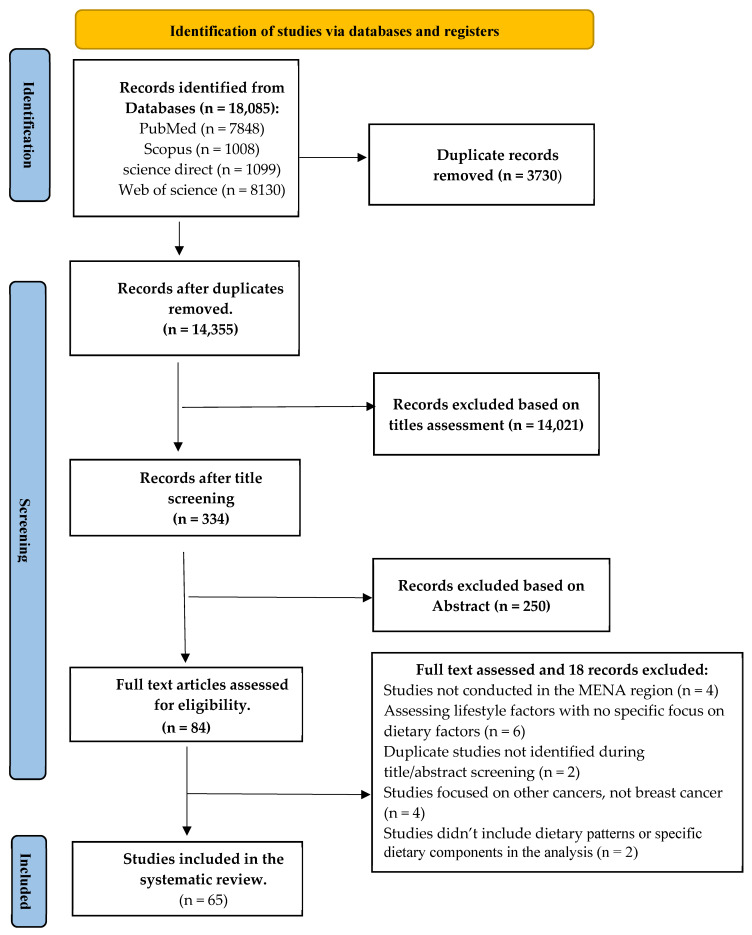
A flow chart illustrating the selection process for studies included in the study.

**Table 1 nutrients-17-00394-t001:** Number of studies exploring dietary factors linked to BC risk.

Dietary Factors	Number of studies
Food and nutrients	Fruit and vegetables	0	8	3
Red meat	2	0	2
Processed meat	1	0	1
Poultry	0	1	2
Dairy products	0	2	4
Milk	4	0	1
Yogurt	0	1	3
Cheese	0	0	3
Cereal	0	1	3
White bread	2	0	0
Whole wheat bread	0	1	0
Fish and seafood	1	3	1
Black tea	0	2	0
Carbohydrate intake	0	0	2
Fat Intake	0	0	3
Calcium Intake	1	1	1
Dietary pattern through priori methods.	MedDiet	0	2	0
Western dietary pattern.	1	0	0
MIND diet	0	0	2
Plant-based diet	0	1	2
Healthy plant-based diet index	0	2	1
Unhealthy plant-based diet index	1	0	2
DASH diet	0	2	2
Dietary insulin index	2	0	0
Dietary insulin load	2	0	0
Dietary Glycemic index	2	0	1
Dietary Glycemic load	0	0	3
Carbohydrates diet	0	1	2
Dietary Diabetes Risk Reduction	0	1	1
Dietary Inflammatory index	6	0	1
Dietary antioxidant	0	2	0
Healthy eating index-2010	0	1	0
Dietary pattern through posteriori methods.	Healthy dietary Pattern	0	3	0
Unhealthy dietary pattern	2	0	1

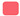
 Increased risk, 
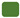
 Decreased risk, 
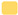
 No significant association.

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
