# Peer review of "Associations Between Dietary Factors and Breast Cancer Risk: A Systematic Review of Evidence from the MENA Region"

_nutrients, 2025, doi:10.3390/nu17030394_

Round 1
Reviewer 1 Report
Comments and Suggestions for Authors
The manuscript entitled “Associations Between Dietary Factors and Breast Cancer Risk: A Systematic Review of Evidence from the MENA Region” discusses the existing evidence regarding the effect of different dietary factors on BC risk among women from the MENA region. The below revisions are recommended:
Abstract: Lines 32-33: 65 studies met the inclusion criteria whereas 66 studies were included. From where does the 66th study come?
In a systematic review, along with detailed information on dietary habits, other factors such as sociodemographic characteristics, weight, height, menstrual and reproductive history, physical activity, and family history of breast cancer among the participants should be included. The current manuscript may be called a review based on dietary factors rather than calling as a systematic review. Please justify.
Page 5: The authors have considered meat consumption as a whole. It is important to mention the type of cooking used for red meat. If red meat is consumed as BBQ (burning of the meat), much more PAHs will be generated and subsequently consumed. PAHs are top-listed carcinogens (visit the US Environmental Protection Agency website). In contrast, if the same red meat is consumed as curry, much less PAHs will be consumed. If reported, the methods of cooking should be included.
Several review articles discussed that the consumption of fruits and vegetables, fish and seafood, and black tea was associated with a reduced BC risk. Please cite some articles.
Page#5: It should be called a Table (not a figure). Please revise.
Please add a discussion on differentiating plant-based diets, healthy plant-based diets, and unhealthy plant-based diets, as mentioned on page 5.
The manuscript contains several grammatical and syntactical errors. It should be thoroughly revised, and the quality of English should be improved.
Comments on the Quality of English LanguagePlease see above.
Author Response
We appreciate reviewer taking the time to review our manuscript and providing us with thoughtful, pertinent feedback. Every comment has been attempted to be addressed and all sections of the paper has been improved accordingly. We trust that we have correctly interpreted all the comments and we hope that the reviewer is satisfied with the changes that have been made to the revised version.t. Please find the detailed responses below and the corresponding revisions/corrections highlighted in track changes in the resubmitted files.
Comments 1: Abstract: Lines 32-33: 65 studies met the inclusion criteria whereas 66 studies were included. From where does the 66th study come? |
Response 1: We thank the reviewer for pointing out this mistake. The studies meeting the inclusion criteria were 65. This mistake has been revised in all the manuscript. |
Comments 2: In a systematic review, along with detailed information on dietary habits, other factors such as sociodemographic characteristics, weight, height, menstrual and reproductive history, physical activity, and family history of breast cancer among the participants should be included. The current manuscript may be called a review based on dietary factors rather than a systematic review. Please justify. |
Response 2: Thank you for your comment. We appreciate the importance of considering multiple factors, such as reproductive and hormonal factors, physical activity, and family history of breast cancer etc., in research addressing breast cancer risk. However, the focus of this review paper was only on dietary factors and their association with breast cancer risk in the MENA region. Globaly, the systematic nature of all reviews is based on the methodology as supported by guidelines such as those of the PRISMA. Additionally, systematic reviews often address a defined research questions to provide clear and more specific conclusions. Expanding the scope to include all potential risk factors for breast cancer may diverge from our initial aim and limiting the paper's ability to provide clear insights into the role of diet in the MENA region. Although our focus was restricted to dietary factors, this targeted approach does not diminish the systematic nature of the review, and this work adheres to the rigorous methodology required for a systematic review. Specifically, we followed established guidelines of PRISMA framework to ensure a transparent, reproducible, and comprehensive approach. To address the reviewer concern, we have updated the manuscript to clarify more the scope and objectives of this paper. Specifically, we have revised the sections of the manuscript to emphasize the review's dietary focus. Also, we have added a discussion point on the importance of other risk factors of breast cancer. |
Comments 3: Page 5: The authors have considered meat consumption. It is important to mention the type of cooking used for red meat. If red meat is consumed as BBQ (burning of the meat), much more PAHs will be generated and subsequently consumed. PAHs are top-listed carcinogens (visit the US Environmental Protection Agency website). In contrast, if the same red meat is consumed as curry, much less PAHs will be consumed. If reported, the methods of cooking should be included. |
Response 3: Thank you for your valuable comment regarding the impact of cooking methods on the production of polycyclic aromatic hydrocarbons (PAHs) and their relevance to the association between red meat consumption and breast cancer risk. We fully acknowledge that cooking methods, particularly high-temperature techniques such as barbecuing or grilling, can significantly influence PAH levels and, consequently, potential carcinogenic effects. Unfortunately, the included studies assessing meat consumption in this systematic review did not provide information on the specific cooking methods used. As a result, we were unable to explore or account for this important variable in our review. This point has been explicitly acknowledged in the discussion section of the revised manuscript. Lines 522-523 We believe this clarification addresses the concern and improves the transparency of the review. |
Comments 4: Several review articles discussed that the consumption of fruits and vegetables, fish and seafood, and black tea was associated with a reduced BC risk. Please cite some articles. |
Response 4: Thank you for your comment. We agree that previous review articles have highlighted the association between the consumption of fruits and vegetables, fish and seafood, and black tea with a reduced risk of breast cancer. We have updated the manuscript to include citations to relevant studies that support these findings. Specifically, we have added the following references in our discussion section:
|
Comments 5: Page#5: It should be called a Table (not a figure). Please revise. |
Response 5: Thank you for your comment. We revised the label and we used "Table" instead of "Figure". |
Comments 6: Please add a discussion on differentiating plant-based diets, healthy plant-based diets, and unhealthy plant-based diets, as mentioned on page 5. |
Response 6: Thank you for your valuable comment. To address this, we have expanded the discussion section to include distinction between plant-based diets, healthy plant-based diets, and unhealthy plant-based diets. We believe this addition enhances the clarity and depth of the manuscript, aligning with your valuable suggestion. Lines 464-475. |
Comments 7: The manuscript contains several grammatical and syntactical errors. It should be thoroughly revised, and the quality of English should be improved. |
Response 7: Thank you for comment. To address this, we have thoroughly revised the manuscript and a professional English editing expert also revised this version of the manuscript. |

Reviewer 2 Report
Comments and Suggestions for Authors
In the current review the authors identified and synthesized the existing evidence regarding the effect of different dietary factors on breast cancer risk among woman from the Middle East and North Africa region. Their findings highlighted that understanding the link between diet and breast cancer within the MENA population is essential for developing effective, population-specific prevention strategies.
Some comments/suggestions:
1.Abstract line 32 and 33 – in the review are included 66 or 65 studies? Please clarify.
2.Introduction, lines 49-50: What about the incidence in 2024? What is estimated for the future? Please complete.3. Table 1 is too long and is difficult to follow. Why did you present the references for which the ”Adjustment for confounding variables” are “NR”? Please split Table 1 in more tables.
4. Point 3.11, you wrote: In Iran, Bidgoli et al. reported that vitamin D supplements were associated with a slightly increased risk of BC”. In my opinion it’s a strange statement because nowadays breast cancer patients are recommended to take vitamin D.
5. For the points 3.2 up to 3.12 add please how many persons/cases are involved in each study.
6. Did you find a difference between the effect of different dietary factors on breast cancer risk among woman /men? Please add.
At exclusion criteria you wrote:” Studies without abstracts or full text were excluded, as were studies that mixed male and female BC results or ….”. You didn’t specify anywhere that you followed only women.
7.Discussion:
- lines 393-94 you wrote that fruits and vegetables rich in vitamin E and beta-carotene inhibit cancer development and at Line 322 you wrote that “vitamins A and E, showed no significant associations”. Please clarify.
- at discussion you have to discuss the results, not to present another studies.
Author Response
We would like to sincerely thank Reviewer 2 for the time and effort spent reviewing our manuscript and for providing constructive feedback. We have carefully addressed each comment and believe that all suggestions have been fully incorporated into the revised manuscript. We trust that Reviewer 2 will find the revisions to be in line with their expectations. Please find the detailed responses below and the corresponding revisions/corrections highlighted in track changes in the resubmitted files.
Point-by-point response to Comments and Suggestions for Authors |
|
|
Comments 1: Abstract line 32 and 33 – in the review are included 66 or 65 studies? Please clarify. |
||
Response 1: We thank the reviewer for pointing out this mistake. The number of studies included was 65 This mistake has been revised in all the manuscript. |
||
Comments 2: Introduction, lines 49-50: What about the incidence in 2024? What is estimated for the future? Please complete. |
||
Response 2: Thank you for your comment. In this version of the manuscript, we have included projections for breast cancer incidence and mortality in the introduction section to provide additional context and have further improved this part of the manuscript accordingly. While we acknowledge the importance of presenting the most up-to-date statistics, the global cancer statistics for 2022 are based on the most recent figures published in 2024 in the Global Cancer Observatory (GCO). Which is a reliable dataset that are widely recognized and used in scientific research due to their robust methodology. At the time of conducting this systematic review, updated cancer incidence data for 2024, were not yet available in the GCO. |
||
Comments 3: Table 1 is too long and is difficult to follow. Why did you present the references for which the” Adjustment for confounding variables” are “NR”? Please split Table 1 in more tables. |
||
Response 3: We appreciate your comment. To address the concern about the table's length, we have split Table 1 into multiple tables according to food groups, nutrients, or dietary patterns. Additionally, the purpose of presenting all studies, including the three references for which the adjusted variables were not clearly described in the main text is to ensure transparency and allows readers to see the complete finding of the evidence available. To ensure transparency, the non-reporting of adjusted variables has also been explicitly discussed as a limitation in the discussion section. We highlight this issue to inform readers that it must be considered when interpreting the findings. We hope this modification improves the clarity of the information. |
||
Comments 4: Point 3.11, you wrote: In Iran, Bigoli et al. reported that vitamin D supplements were associated with a slightly increased risk of BC”. In my opinion it’s a strange statement because nowadays breast cancer patients are recommended to take vitamin D. |
||
Response 4: Thank you for your comment. To clarify, the statement was made in the results section where we presented the findings of the included studies as reported, without making any judgment. Specifically, the result you referred to comes from a study by Bigoli et al. conducted in Iran, which had a small sample size of 176 participants. This study found a modest association between vitamin D supplementation and a slightly increased risk of breast cancer in this population. However, it's important to note that this does not imply a generalizable link between vitamin D and breast cancer risk. The study was observational and conducted in a specific context, and the association found was modest and potentially influenced by confounding factors. As we acknowledged that in the discussion section as a limitation of observational studies. |
||
Comments 5: For the points 3.2 up to 3.12 add please how many persons/cases are involved in each study. |
||
Response 5: The number of participants or cases involved in each study have been added. Thank you for picking up on this information. |
||
Comments 6: Did you find a difference between the effect of different dietary factors on breast cancer risk among woman /men? Please add. |
||
Response 6: Thank you for your thoughtful comment. In our systematic review, we focused on studies assessing the association between dietary factors and breast cancer risk among women as it was mentioned in section "2.3. Inclusion and exclusion criteria". This focus was due to the limited studies investigating dietary factors and breast cancer risk among men. Consequently, our analysis did not include data comparing the effects of dietary factors on breast cancer risk between women and men. |
||
Comments 7: At exclusion criteria you wrote:” Studies without abstracts or full text were excluded, as were studies that mixed male and female BC results or ….”. You didn’t specify anywhere that you followed only women. |
||
Response 7: Thank you for pointing on this point. We would like to confirm that this focus on studies conducted among women has already been explicitly stated in the "2.3. Inclusion and exclusion criteria" section in Methods. Specifically, we mentioned the following: "Articles were eligible for inclusion if they met the following criteria: they reported findings from primary research studies conducted among women with BC in the MENA region, assessed dietary factors influencing BC risk." In this revised version we replace women by females BC. |
||
Comments 8-a: Discussion: lines 393-94 you wrote that fruits and vegetables rich in vitamin E and beta-carotene inhibit cancer development and at Line 322 you wrote that “vitamins A and E, showed no significant associations”. Please clarify. |
||
Response 8-a: Thank you for your observation. We appreciate your careful reading of the manuscript and will clarify the context of these two statements.
|
||
Comments 8-b: Discussion: at discussion you have to discuss the results, not to present another study. |
||
Response 8-b: Thank you for your feedback. We appreciate your point regarding the focus of the "Discussion" section. In response to your comment, we have revised and restructured the "Discussion" to prioritize the interpretation of our findings, drawing clearer comparisons with previous established evidence, included more critical analysis of how our results contribute to understanding dietary factors in breast cancer risk in MENA. We believe these changes address your concerns. Thank you again for your valuable feedback. |
